statistics/differential equations/biomathematics

tumour cells, cancer invasion, approximate Bayesian computation, Bhattacharyya distance, gradient matching, generalized additive models

**Author for correspondence:**
Yunchen Xiao
e-mail: ycx@st-andrews.ac.uk

# Calibrating models of cancer invasion: parameter estimation using approximate Bayesian computation and gradient matching

Yunchen Xiao, Len Thomas and Mark A. J. Chaplain

School of Mathematics and Statistics, University of St Andrews, St Andrews, KY16 9SS UK

YX, 0000-0002-1297-7439; LT, 0000-0002-7436-067X;
MAJC, 0000-0001-5727-2160

We present two different methods to estimate parameters within a partial differential equation model of cancer invasion. The model describes the spatio-temporal evolution of three variables—tumour cell density, extracellular matrix density and matrix degrading enzyme concentration—in a one-dimensional tissue domain. The first method is a likelihood-free approach associated with approximate Bayesian computation; the second is a two-stage gradient matching method based on smoothing the data with a generalized additive model (GAM) and matching gradients from the GAM to those from the model. Both methods performed well on simulated data. To increase realism, additionally we tested the gradient matching scheme with simulated measurement error and found that the ability to estimate some model parameters deteriorated rapidly as measurement error increased.

# 1. Introduction

Systems of differential equations are used frequently to model and predict the behaviour of dynamical systems. Application areas include physics [1], engineering [2], ecology [3] and medicine [4]. A common issue is that some or all model parameters are not known, and one needs to calibrate the model based on available observed data, i.e. find model parameters that give the closest fit to a set of observations on the system. This process is also known as parameter estimation or solving the inverse problem. Here, we develop two calibration schemes: one inspired by approximate Bayesian computation (ABC) and the other built upon the idea of gradient matching. We illustrate their application to a partial differential equation (PDE) model

for cancer invasion and metastasis, based on synthetic data. Since they model the evolution in both time and space of multiple variables, systems of PDEs can be more challenging to analyse than their purely temporal counterparts—ordinary differential equations, ODEs—and as such have received less attention to date.

For almost all differential equation models, a closed-form solution to provide the parameter values that give the best fit to the observed data is not available, and so an iterative scheme is used. Classically, this involves solving the model system for a candidate set of parameter values and comparing this solution with the observations; the candidate set is then updated in a way that tends to produce a better solution and the process is repeated to convergence [5,6]. One standard metric of fit is the sum of squared differences between model solution and observed values. Under the assumption that the observations arise independently from a normal distribution then the least-squares parameter estimates are maximum-likelihood estimates. This allows quantification of uncertainty in the estimates, for example by normal confidence intervals. Early texts suggesting this 'nonlinear least squares' approach include [7–9]. One issue is that the likelihood surface may be multimodal, making it difficult to ensure the globally best parameter values are found [10] .

Bayesian methods have also been employed to draw inferences about differential equation models—this allows for a richer characterization of uncertainty and incorporation of prior information about model parameters. Inference is often via Markov chain Monte Carlo (MCMC) (e.g. [11]), and an R package, dbInfer, has been developed to make this type of modelling more widely accessible [12,13]. More complex algorithms have been developed to improve estimation performance—for example, Pǎun *et al*. [14] proposed three MCMC-related algorithms and also used constrained optimization to obtain good starting values for the Markov chains. Alternatively, approximate Bayesian computation (ABC) has been proposed [15]—this bypasses the need to give an explicit form for the likelihood. We are aware of only one application of ABC to PDE models [16].

One strong disadvantage of all of the above approaches is that they require solving the differential equation model at each step of the inference algorithm. This can make them prohibitively slow for more complex models or large datasets. An alternative approach, first proposed by Varah [17], is gradient matching. Here, instead of numerically integrating the model and comparing the solution to the data, a smooth interpolant is fitted to the data, and gradients obtained from this interpolant are compared with the gradients predicted by the model. The model parameters are adjusted until the predicted gradients best match those from the interpolant. No numerical integration is required, and the smooth interpolant fitting only needs to occur once—thereafter, comparing model-predicted gradients to interpolant gradients is computationally cheap. In general, the gradient matching methods can be classified into those adopting a traditional 'two-step approach' as explained above, where the interpolants exert a unidirectional influence on the ODEs (e.g. [17,18]), and those that perform the smoothing and gradient matching at the same time, allowing the ODEs to exert an influence on the interpolants. Two main examples of the latter are approaches based on the use of reproducing kernel Hilbert spaces (RKHS) [19–21] and those applying Gaussian processes [18,22,23].

Here, we develop two calibration schemes, one based on ABC and the other on two-stage gradient matching, and apply them to a PDE model of cancer invasion and metastasis. Our focus is on accurate estimation of model parameters when applied to data simulated from the model—a necessary first step for reliable inference. We leave the two important real-world issues of uncertainty quantification and model assessment (goodness-of-fit) for future work.

In the following paragraphs, we provide a brief overview of the biological processes underlying the PDE model.

## 1.1. Solid tumour growth and spread

The establishment and development of a primary solid tumour usually begins with a single normal cell being transformed as a result of mutation(s) in certain key genes. The transformed cells can escape from the body's homeostatic mechanisms, leading to inappropriate proliferation [24] and the formation of a cluster (nodule) of tumour cells. The nodule can expand to an avascular tumour consisting of approximately $10^6$ cells, with a diameter up to approximately 0.1–0.2 cm [24]. Avascular tumours must initiate angiogenesis—the recruitment of blood vessels for further growth. The tumour cells secrete tumour angiogenic factors (TAFs) to induce endothelial cells in neighbouring blood vessels to degrade their basal lamina and migrate towards the avascular tumour [24]. The newly formed vessels eventually develop a capillary network that connects with the tumour and vascularizes it. With the

presence of this capillary network, the process of angiogenesis is complete, and necessary nutrients for further growth will be supplied to the tumour. Tumour cells can also find their way into the circulation and be deposited in secondary sites in the body, resulting in metastasis [24]. Metastatic spread is responsible for around 90% of the deaths from cancer [25].

A prominent part of the invasive/metastatic process is the ability of the cancer cells to degrade the surrounding tissue or extracellular matrix (ECM), mediated by a number of matrix degrading enzymes (MDEs) [24,26–28]. ECM is a three-dimensional complex network of macromolecules, such as collagens, laminin, fibronectin, glycoproteins and vitronectin. A number of MDEs such as urokinase plasminogen activator (uPA) and matrix metalloproteinases (MMPs) have been described [24]. Regulation of matrix-degrading activity is highly complex; no MDE is completely specific for one element of the ECM, and both uPA and MMPs have played a role in the necessary steps of tumour metastasis [24,28–38].

Over the past few decades, many mathematical models related to cancer invasion and metastasis have been proposed by different authors [39]. The computational simulations of these models exhibit travelling-wave-like behaviour of the cancer cells and other biologically relevant variables. Here, we focus on a representative model proposed by Anderson *et al.* [24] as our subject of investigation, which is a continuum deterministic model based on a system of reaction–diffusion-taxis equations, describing the (spatio-temporal) evolution of three key variables involved in tumour cell invasion: tumour cells, ECM and MDEs.

## 2. Methods

### 2.1. PDE system and its non-dimensionalizations

The PDE model we investigate is based on the cancer invasion and metastasis model proposed by Anderson *et al.* [24]. It is described by the following set of equations:

$$
\left.
\begin{aligned}
\frac{\partial n}{\partial t} &= D_n \frac{\partial^2 n}{\partial x^2} - \chi \frac{\partial}{\partial x}\left(n \frac{\partial f}{\partial x}\right) + R_n n\left(1 - \frac{n}{n_0} - \frac{f}{f_0}\right), \\
\frac{\partial f}{\partial t} &= -\delta m f, \\
\frac{\partial m}{\partial t} &= D_m \frac{\partial^2 m}{\partial x^2} + \mu n - \lambda m,
\end{aligned}
\right\}
\tag{2.1}
$$

where $n$ is the tumour cell density, $f$ is the ECM density, $m$ is the MDE concentration, $t$ is time in seconds and $x$ is distance from tumour centre in cm. The interpretation of these equations is as follows. The first equation models the profile of tumour cells density: the first term is Fickian diffusion (random motility), where $D_n$ is the diffusion coefficient in $\text{cm}^2\,\text{s}^{-1}$, the second models the process of haptotaxis (the directed migratory response of tumour cells to gradients of ECM), where $\chi$ is the haptotactic coefficient in $\text{cm}^2\,\text{s}^{-1}\,\text{M}^{-1}$, and the third models logistic growth of tumour cells; $n_0$, $f_0$, $m_0$ are reference densities or concentrations. This third term did not appear in the model of Anderson *et al.* but was added here to represent competition of resources/space between tumour cells and the ECM; $R_n$ represents the logistic growth rate, measured in $\text{s}^{-1}$. The second equation models the profile of the ECM: the only biological phenomenon involved is its degradation by MDE; $\delta$ represents the degradation rate, measured in $\text{s}^{-1}\,\text{M}^{-1}$. The third equation models the profile of the MDE: the first term is random motility, where $D_m$ has the same unit as $D_n$, $\text{cm}^2\,\text{s}^{-1}$, the second is the production of MDE by tumour cells, where $\mu$ represents the growth rate measured in $\text{s}^{-1}$, and the third models chemical (and other) decay of MDE, where $\lambda$ is the decay rate measured in $\text{s}^{-1}$.

In order to solve this PDE system numerically, all three equations above were non-dimensionalized. We did this in the standard manner, as follows. First, we rescaled the distance with an appropriate length scale $L$. Given that maximum invasion distance of cancer cells at the early stage of invasion varies from 0.1 to 1 cm, we took $L = 1$ cm [40]. Then, we rescaled the time with $\tau = L^2/D$, where $D$ is a reference chemical diffusion coefficient $\sim 10^{-6}\,\text{cm}^2\,\text{s}^{-1}$ [40]. Last, we rescaled the tumour cell density with $n_0$, ECM density with $f_0$ and MDE concentration with $m_0$. These rescalings define the following variables:

$$
\tilde{n} = \frac{n}{n_0}, \quad \tilde{f} = \frac{f}{f_0}, \quad \tilde{m} = \frac{m}{m_0}, \quad \tilde{x} = \frac{x}{L} \quad \text{and} \quad \tilde{t} = \frac{t}{\tau}.
\tag{2.2}
$$

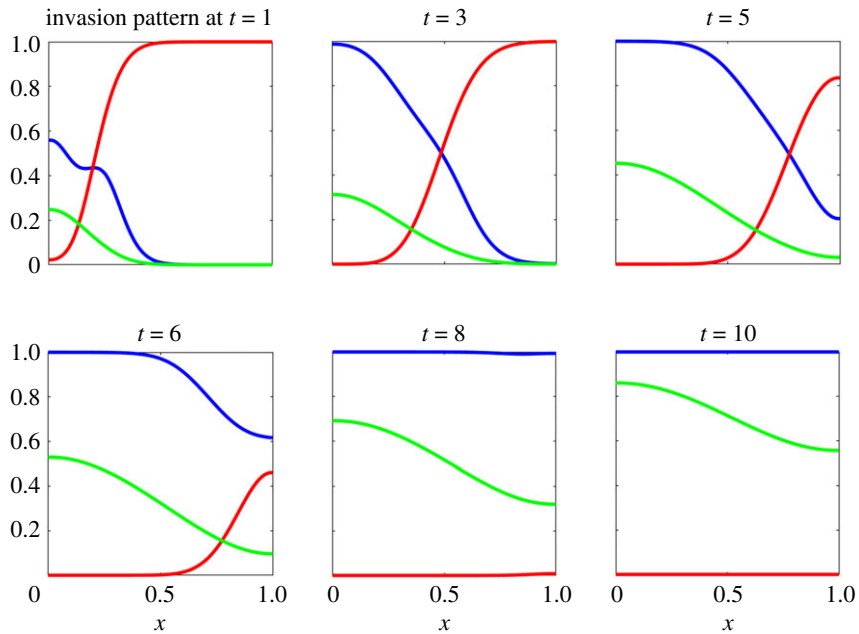

**Figure 1.** Reference invasion pattern at times $t = 1, 3, 5, 6, 8$ and $10$. Tumour cell density is shown as a blue curve, ECM density as a red curve and MDE concentration as a green curve.

Dropping all the tildes for notational convenience, the dimensionless system is

$$\frac{\partial n}{\partial t} = d_n \frac{\partial^2 n}{\partial x^2} - \gamma \frac{\partial}{\partial x}\left(n\frac{\partial f}{\partial x}\right) + r_n n(1 - n - f),$$

$$\frac{\partial f}{\partial t} = -\eta m f$$

and

$$\frac{\partial m}{\partial t} = d_m \frac{\partial^2 m}{\partial x^2} + \alpha n - \beta m, \tag{2.3}$$

where $d_n = D_n/D$, $\gamma = \chi f_0/D$, $r_n = R_n\tau$, $\eta = \tau m_0\delta$, $d_m = D_m/D$, $\alpha = \tau\mu n_0/m_0$ and $\beta = \tau\lambda$.

## 2.2. Reference datasets

To demonstrate the performance of the calibration schemes, we generated synthetic datasets from the PDE model under a single set of chosen 'reference' parameter values. Considering each of these as the observed data, we investigated how accurate the parameter values estimated by our schemes are in relation to the reference values. The following dimensionless parameter values were used: $d_n = 0.01$, $\gamma = 0.05$, $\eta = 10$, $d_m = 0.01$, $\alpha = 0.1$, $r_n = 5$ and $\beta = 0$. Setting $\beta$ to zero means that decay in MDE is negligible during the time scale of the observations, as was done by Anderson *et al.* [24]; we make this an assumption of the model fitting and exclude this parameter from estimation.

The simulations used a finite-difference scheme. In the dimensionless form, we considered the one-dimensional spatial domain [0,1], which was discretized into 80 points; the step size for space was thus $\Delta x = 1/(80 - 1) \approx 0.0127$. The step size for time in the finite difference scheme was set to be 0.001. For each variable, the evolution of its simulated pattern was recorded at 11 time points, $t = 0, 1, 2, \ldots, 10$.

The resulting invasion pattern is shown in figure 1. Tumour cells can be seen to invade into the surrounding ECM, simulating the production of MDE which results in the complete degradation of ECM in the domain.

The reference dataset therefore consisted of all three variables (tumour cells density $n$, ECM density $f$ and MDE concentration $m$) measured over 80 evenly spaced points in the one-dimensional spatial domain between 0 and 1, and at 11 evenly spaced time points between 0 and 10. For this dataset (referred to in the Results as the 'main reference dataset'), measurements were assumed to be recorded without error. This dataset was used to demonstrate both the ABC-related and the gradient matching schemes.

While this reference dataset is useful for an initial demonstration, it is unrealistic in the quantity of measurements ($80 \times 11 = 880$) taken on each variable, the fact that all three variables are measured,

and the lack of measurement error. A proper treatment of all three issues is beyond the scope of this paper and is something we return to in the Discussion. However, we do go some way to address the measurement error issue.

To simulate measurement error, we generated 200 reference datasets where each data point was drawn from a gamma distribution with mean equal to its true value and a specified coefficient of variation (CV). The reason to generate multiple datasets in this instance is that we do not expect the exactly correct parameter values to be recovered from each dataset when the data are measured with error—however, we do expect that if the measurements are unbiased then the correct parameter values will be recovered on average over multiple replicate datasets. Because of the computational burden of fitting multiple datasets, we were not able to perform this check for the ABC-related scheme, and hence only fitted the measurement error datasets using the gradient matching scheme. The CV values used were 0.01, 0.025, 0.05, 0.075, 0.10.

Lastly, as a check of the generality of our conclusions about the ABC-related scheme, we generated two additional reference datasets with different, but still realistic, parameter values (and no measurement error). The values used are shown in electronic supplementary material, appendix B (table S5).

All simulations and associated fitting, using the schemes described below, were performed in the R statistical software [41].

## 2.3. ABC-related optimization scheme

### 2.3.1. ABC to PDE

A general outline of our approach applying ABC to PDE models is as follows.

(i) Assign a uniform initial distribution to each parameter to be estimated. Simulate multiple sets of parameter values by sampling from their corresponding initial distributions.
(ii) For each set of simulated parameter values, substitute them into the PDE solver to obtain simulated values for tumour cell density, ECM density and MDE concentration over space and time.
(iii) Compare each simulation with the reference dataset using a chosen discrepancy measuring metric, which in turn uses summary statistics calculated from each simulated dataset and the reference dataset.
(iv) Use the discrepancy values to assign a resampling probability to each set of parameter values such that the smaller the discrepancy value, the higher the resampling probability. Resample the parameter sets with replacement, based on the resampling probability.
(v) Add small perturbations to all parameter values.
(vi) Repeat steps (ii) to (v) for a fixed number of iterations or until the average discrepancy is reduced to a satisfactory level.
(vi) Take the sample mean of the values for each parameter as the final estimate.

In the ABC literature, the initial distribution is viewed as a prior distribution and the sets of parameter values obtained at step (vi) are considered to be increasingly accurate estimates of the parameters' posterior distribution. However, in our case, with no stochasticity in the model and no measurement error in the reference data, the algorithm will eventually converge on the best fitting values with no variation between parameter sets beyond that added at step (v). Hence, we consider our scheme to be a stochastic optimization scheme, suitable for estimating parameter values but not for assessing uncertainty on the estimates.

### 2.3.2. Discrepancy measure: Bhattacharyya distance

Central to any ABC-related scheme is selection of suitable summary statistics and a corresponding discrepancy measure. Here, we use summary statistics based on the mean and variance of each variable (tumour cell density, ECM density and MDE concentration) across time at fixed points in space, and a discrepancy measure based on a standard measure of distance between two probability distributions, the Bhattacharyya distance [42].

For univariate samples from a normal distribution, the Bhattacharyya distance is given by

$$BC_{1,2} = \frac{1}{4}\ln\left[\frac{1}{4}\left(\frac{\sigma_1^2}{\sigma_2^2} + \frac{\sigma_2^2}{\sigma_1^2} + 2\right)\right] + \frac{1}{4}\left[\frac{(\mu_1 - \mu_2)^2}{\sigma_1^2 + \sigma_2^2}\right], \tag{2.4}$$

**Table 1.** Initial distributions used in the ABC-BCD scheme. Justification for these values is given in electronic supplementary material, appendix A. U(l, u) denotes a uniform distribution with lower and upper bounds l and u.

| parameter | initial distribution | mean (s.d.) |
|---|---|---|
| $d_n$ | $U(6.90 \times 10^{-5}, 2.00 \times 10^{-2})$ | $1.00 \times 10^{-2}$ $(5.75 \times 10^{-3})$ |
| $\gamma$ | $U(5.00 \times 10^{-3}, 2.60 \times 10^{-1})$ | $1.33 \times 10^{-1}$ $(7.36 \times 10^{-2})$ |
| $r_n$ | $U(3.50, 9.00)$ | $6.25$ $(1.59)$ |
| $\eta$ | $U(7.00, 1.80 \times 10)$ | $1.25 \times 10$ $(3.18)$ |
| $d_m$ | $U(1.00 \times 10^{-4}, 3.30 \times 10^{-2})$ | $1.66 \times 10^{-2}$ $(9.50 \times 10^{-3})$ |
| $\alpha$ | $U(7.00 \times 10^{-2}, 1.80 \times 10^{-1})$ | $1.25 \times 10^{-1}$ $(3.18 \times 10^{-2})$ |

where $\mu_i$ and $\sigma_i^2$ are the mean and variance of the $i$th sample. For each variable ($n$, $f$ and $m$) we used the above form of the Bhattacharyya distance between simulated and reference data at each location in space over the 10 time points ($t = 0$ not included as this is the initial condition, assumed known). Overall discrepancy for each variable was calculated by summing over space points

$$\rho_y = \sum_{i=1}^{80} BC_{\text{sim,ref}}(y, x_i), \tag{2.5}$$

where $y$ is one of the three variables $n$, $f$ or $m$ and $BC_{\text{sim,ref}}(y, x_i)$ is the Bhattacharyya distance between simulated and reference data calculated for variable $y$ at location $x_i$. Total discrepancy, where needed, was calculated by summing across variables: $\rho = \rho_n + \rho_f + \rho_m$.

For our scheme to converge to the true parameter values, there is an implicit assumption that a Bhattacharyya distance of zero means that the data perfectly match model predictions. This assumption would fail if it were possible for the model to generate a dataset with the same means and variances as the data, but from a different set of parameter values. This is unlikely in our case study; in a real-world application, where the model is necessarily an approximation to the real-world data-generating process and the data also contain measurement errors, it would be required that smaller values of the discrepancy measure correspond to better values of the model parameters.

### 2.3.3. ABC-BCD (approximate Bayesian computation–Bhattacharyya distance) optimization scheme

Our ABC-BCD optimization scheme proceeds as follows.

  (i) Identify all the unknown parameters within the PDE system, $P = \{p_1, p_2, \ldots, p_m\}$. Assign an initial distribution to each parameter. The initial distributions we chose for the parameters are shown in table 1, and justification for these distributions are given in electronic supplementary material, appendix A.
 (ii) Identify the equation in the system that contains the lowest number of parameters. (We started with the ECM density profile, as it has only one parameter to be estimated, then we moved on to MDE concentration profile (two parameters). Finally, we investigated the tumour cells density profile (three parameters).)
(iii) Now set the round indicator $j$ to be 1.
   (a) Sample $K$ sets of parameter values from the initial distributions. Note that $K$ and round indicator $j$ are independent in this study.
   (b) For each set of simulated parameter values, substitute them back to the PDE solver to obtain corresponding values for tumour cell density, ECM density and MDE concentration over space and time.
   (c) For each simulation, use the Bhattacharyya distance formula in equation (2.4) to obtain the discrepancy between the simulated data and the reference data; denote this result as $\rho_{yi}^j$, $i = 1, \ldots, K$. Note that we only calculate the discrepancies among the variables in the current equation and the equations that have been evaluated previously. When evaluating the ECM density profile, this is just $\rho_{fi}^j$. Then we move on to evaluate the MDE concentration profile, we calculate $\rho_{mi}^j + \rho_{fi}^j$. Lastly, while evaluating the tumour cells density profile, we calculate the total discrepancy $\rho_{ni}^j + \rho_{fi}^j + \rho_{mi}^j$.
   (d) Calculate the average (mean) discrepancy of the parameters in the current round. Check if the distance (difference in absolute value) between the average discrepancy of the parameters

in the current round and that of the parameters in round 1 has reached a threshold value, i.e. $|\bar{\rho}_y^j - \bar{\rho}_y^1| < \epsilon\ \bar{\rho}_y^1$, where $\varepsilon$ is a positive number less than 1. If no:

i. Convert the discrepancy values to resampling weights. First calculate $w_i^* = \rho_{yi}^{-t}$ for $i = 1, \dots K$, where $t$ is a positive integer increased by 50% in every subsequent round. Then rescale the weights in the standard manner

$$w_i = \frac{w_i^*}{\sum_i^K w_i^*}.$$

In some simulations, there are computational singularities resulting in undefined real values; in these cases, the corresponding weights are set to 0.

ii. Resample another $K$ sets of parameter values with replacement, with probability equal to the resampling weights obtained in the previous step.

iii. Add a small perturbation to the values for each parameter. The following procedure retains the mean and variance of each parameter. Let $p_{li}$ be the value of the $l$-th parameter in the $i$-th parameter set and $\bar{p}_l$ be the sample mean over the $K$ parameter sets. Calculate the parameter sample variance as $S_l^2 = \frac{1}{K-1}\sum_{i=1}^K (p_{li} - \bar{p}_l)^2$. Generate a new value for $p_{li}$ by sampling a random number from $N((\sqrt{1-h^2})p_{li} + (1 - \sqrt{1-h^2}))\bar{p}_l, h^2 S_l^2)$, where $N(\mu, \sigma^2)$ is a normal distribution with mean $\mu$ and variance $\sigma^2$, and $h$ is a relatively small number [43].

iv. $j = j + 1$, record the perturbed parameter values obtained in the previous step, go to (iii)(b) and proceed.

Else:

i. Terminate evaluations of the current equation. For the parameters in the equation(s) we just finished evaluating, record their values from the parameter sets in the current round.

ii. Proceed to the next equation, sample another $K$ sets of parameter values:

A. For the parameters in the equations that have not been evaluated yet, sample their values from the initial distributions.

B. For the parameters in the equations that have been evaluated previously, adopt their values from (iii)(d)i.

iii. With the newly formed $K$ sets of parameter values, go to (iii) and proceed, until the final samples for all parameters in the PDE model are obtained.

(iv) After the final samples of all parameters are obtained, we take the means of these samples to be the estimated parameter values that can give the best fit to the synthetic data. (In the fully Bayesian setting, this would be the 'posterior mean'—a Bayes estimator that has a quadratic loss function.)

The optimization scheme is stochastic, with the amount of Monte Carlo error controlled by the number of samples, $K$. Here, we used $K = 10\,000$. We checked the Monte Carlo error by undertaking two additional runs on the same reference dataset and computing the standard error of the parameter estimates across the three runs. Overall accuracy is governed by the stopping criterion $|\bar{\rho}_y^j - \bar{\rho}_y^1| < \epsilon\ \bar{\rho}_y^1$ and the increasing bandwidth $t$ in weight calculations. Different $\epsilon$'s were used in the evaluations of the three different density profiles, as follows. It was set to be 0.8 when evaluating the ECM density profile alone, to prevent particle depletion in early rounds when only one parameter was being estimated. We then raised it to 0.9 in the evaluations of ECM and MDE profiles. Finally, it was set to be 0.98 when all density profiles are being evaluated. In our opinion, particle depletion is a concerning issue in early rounds, but should not cause any troubles in the last few rounds since our goal is to obtain accurate parameter point estimates. The bandwidth of weights $t$ was chosen to start from 0.5 and increased by 50% in every subsequent round, it was reset to 0.5 when we proceeded to evaluate the next equation. By setting such stopping criterion and adaptive bandwidths for weights, the resampling surface became steeper in later rounds, parameter sets had minor Bhattacharyya distances to the reference values could then be resampled with heavier weights and convergence to the true values could be guaranteed. For the perturbation value $h$ in (iii)(d)iii, we chose $h = 0.05$. Lastly, as a further diagnostic of the scheme, we ran it on the two additional reference datasets that had been generated using different true parameter values (electronic supplementary material, table S4).

Our ABC scheme has both similarities with and differences from the approximate Bayesian computation–sequential Monte Carlo (ABC-SMC) scheme described in Toni *et al.* [15]. Both schemes

use the idea of weighted resampling and particle perturbations to derive the parameters to be evaluated in the next round from the ones being evaluated in the current round. The main differences between our scheme and the traditional ABC-SMC scheme are:

(i) The discrepancy between the summary statistics and the computation of weights are two separate processes in the traditional ABC-SMC scheme, while we combined them into one in our scheme. If $t$ rounds are carried out in total, the traditional ABC-SMC scheme usually sets up a decreasing sequence of tolerance levels $\varepsilon_1, \varepsilon_2, \ldots, \varepsilon_t$ at the beginning. After round 1, the parameters to be evaluated in the next round are resampled and perturbed from the ones being evaluated in the current round using weights associated with parameter densities; the weights here act as a first filter. Then the tolerance levels are introduced as a second filter: the simulated and reference summary statistics are compared using certain discrepancy measuring metric, and a perturbed parameter set is accepted only if the discrepancy between its simulated summary statistics and the reference ones is less than the tolerance level of the current round. By contrast, in our scheme, we incorporated the discrepancy measurements into the computation of weights, the weights are then used as the only filter to resample the parameters and ensure convergence.

(ii) Due to the absence of observation errors in our ABC application, a rigorous analysis of parameter densities was not undertaken in our work. Therefore, the weights of parameter sets in our scheme solely depend on the discrepancies between summary statistics. In the traditional ABC-SMC scheme, weights of parameter sets are often related to parameter densities and help to derive the joint and marginal posterior densities. Thus, for readers who wish to draw posterior inferences, it is necessary to incorporate observation errors in the step of data simulation and parameter densities in the step of weight calculations.

## 2.4. Gradient matching scheme

Fitting was performed using the R statistical software [41]. For each variable in the PDE model ($n$, $f$ and $m$), the measurements were modelled using a separate generalized additive model (GAM) [44], with space and time as explanatory variables. The explanatory variables were entered as a two-dimensional adaptive P-spline (via the option `bs = 'ad'` in the `gam` function of the `mgcv` package)—this allowed for the amount of smoothness in the fitted surface to vary across space and time. The response variables ($n$, $f$ and $m$) were assumed to be gamma-distributed and a log link function was used.

Given the fitted GAMs, we then calculated the temporal gradients on the left-hand side ($\partial n/\partial t$, $\partial f/\partial t$, $\partial m/\partial t$) and the spatial gradients on the right-hand side of the PDE system (denote them as $N(t, x)$, $F(t, x)$ and $M(t, x)$) using the fitted values from the GAMs. The discrepancies between the two sides of the equations were then calculated as the sum of squared differences between left and right sides of each equation, evaluated over a grid of space and time points

$$G_n = \sum_{i=1}^{9} \sum_{j=1}^{78} \left( \frac{\partial n}{\partial t}\Big|_{t=i, x=j\times h} - N(t, x)\Big|_{t=i, x=j\times h} \right)^2$$

$$G_f = \sum_{i=1}^{9} \sum_{j=1}^{78} \left( \frac{\partial f}{\partial t}\Big|_{t=i, x=j\times h} - F(t, x)\Big|_{t=i, x=j\times h} \right)^2$$

and
$$G_m = \sum_{i=1}^{9} \sum_{j=1}^{78} \left( \frac{\partial m}{\partial t}\Big|_{t=i, x=j\times h} - M(t, x)\Big|_{t=i, x=j\times h} \right)^2. \tag{2.6}$$

Note that we excluded the boundary points of time ($t = 0.0$ and $t = 10.0$) and space ($x = 0$ and $x = 1$) from our comparisons, resulting in a summation of $78 \times 9$ terms for each variable. Let $\underline{\theta}$ denote the vector of all the parameters within the PDE system, $\underline{\theta} = \{d_n, \gamma, r_n, \eta, d_m, \alpha\}$, the estimated parameter values which gave the best fit to the noisy data were obtained by minimizing (using the `optim` function with the default 'Nelder-Mead' method in R) the objective function

$$G(x, t, n, f, m, \underline{\theta}) = G_n + G_f + G_m. \tag{2.7}$$

Starting values for the minimization algorithm were the means of the initial distributions derived previously for the ABC-BCD scheme (table 1).

In addition to using this scheme on the main reference dataset, we also evaluated its performance when measurement error is introduced to the observations. As stated earlier, 200 datasets were

**Table 2.** Parameter estimates from the ABC-related and gradient matching (GM) schemes fitted to reference data with no measurement error. (See electronic supplementary material, appendix B (a) for estimates at the end of each round of the ABC-related scheme.)

| parameter | reference values | estimates | | percentage error | | mean squared error | |
|---|---|---|---|---|---|---|---|
| | | ABC | GM | ABC | GM | ABC | GM |
| $d_n$ | $1.00 \times 10^{-2}$ | $1.02 \times 10^{-2}$ | $9.96 \times 10^{-3}$ | 2.31 | $-4.28 \times 10^{-1}$ | $1.37 \times 10^{-5}$ | — |
| $\gamma$ | $5.00 \times 10^{-2}$ | $5.18 \times 10^{-2}$ | $4.57 \times 10^{-2}$ | 3.70 | $-8.67$ | $1.43 \times 10^{-4}$ | — |
| $r_n$ | 5.00 | 5.23 | 4.60 | 4.63 | $-8.10$ | $8.54 \times 10^{-1}$ | — |
| $\eta$ | $1.00 \times 10^{-1}$ | 9.91 | $1.03 \times 10$ | $-9.21 \times 10^{-1}$ | 3.06 | $4.61 \times 10^{-2}$ | — |
| $d_m$ | $1.00 \times 10^{-2}$ | $1.04 \times 10^{-2}$ | $9.53 \times 10^{-3}$ | 3.75 | $-4.72$ | $4.82 \times 10^{-7}$ | — |
| $\alpha$ | $1.00 \times 10^{-1}$ | $9.97 \times 10^{-2}$ | $9.93 \times 10^{-2}$ | $-2.53 \times 10^{-1}$ | $-7.49 \times 10^{-1}$ | $8.07 \times 10^{-3}$ | — |

generated at each of five levels of measurement CV, ranging from 0.01 to 0.1. The gradient matching scheme was used to estimate parameter values for each of these datasets, and the mean estimated value was calculated for each parameter at each CV level. Comparing the mean estimated value to the true values enabled us to evaluate bias in parameter estimates.

Our initial runs showed rapidly increasing bias with increasing CV for some parameters, and we speculated that this may be due to difficulties estimating parameters associated with the more numerically complex terms in the model—i.e. those associated with second-order derivatives and haptotaxis. We therefore repeated the above exercise three times: fixing both second-order terms $d_n$ and $d_m$, fixing both complex terms in the tumour cell density equation $d_n$ and $\gamma$ and fixing all the parameters in the tumour cell density equation at their true values to see if any improvements can be made.

# 3. Results

## 3.1. Performance on datasets with no measurement error

Parameter estimates obtained from the two schemes on the main reference dataset are given in table 2. In general, they were very close to the true values used to generate the data, with absolute percentage error less than 9% for all parameters. Errors were noticeably higher for the parameters $d_n$, $\gamma$ and $r_n$, all of which are part of the first and most complex equation governing the change in tumour cell density. Overall, the ABC-related scheme performed slightly better, with lower percentage error on almost all parameters. Plots of initial and final densities for the parameters under this scheme are shown in figure 2, a figure of pairwise heat maps of these final densities is shown in figure 3. As would be expected given the better parameter estimates, the ABC-related scheme also produced a slightly closer reconstruction of the correct solution to the PDEs (see electronic supplementary material, appendix B (d)).

The two additional runs of the ABC-related scheme on the same reference dataset produced very similar results, and the estimated Monte Carlo error was less than 4.5% for all parameters (electronic supplementary material, table S4).

The ABC-related scheme was also run on two additional reference datasets with different parameter values. The percentage error was less than 6.5% for all parameters (electronic supplementary material, table S5), confirming that this scheme can reliably retrieve model parameters in synthetic data.

## 3.2. Performance of gradient matching scheme on datasets with measurement error

Parameter estimates obtained from the gradient matching scheme under different levels of measurement error are given in figure 4. Estimation accuracy for the tumour cell-related parameters fell rapidly with increasing perturbation level: at CV of 0.1, the mean estimate of $d_n$, $\gamma$ and $r_n$ deviated 60.6%, 45.2% and 39.7%, respectively, from their true values. The behaviour of the other parameters was better: mean estimates of $\eta$, $d_m$ and $\alpha$ were 3.52%, 16.6% and 1.87% from their true value.

Three sensitivity tests were carried out to investigate whether fixing the unstable parameters at their true values can improve the accuracy of other parameters. Results are shown in figure 4. In the first test, the two diffusion coefficients $d_n$ and $d_m$ were fixed; bias increased in $r_n$ and $\eta$ and decreased in $\gamma$. In the

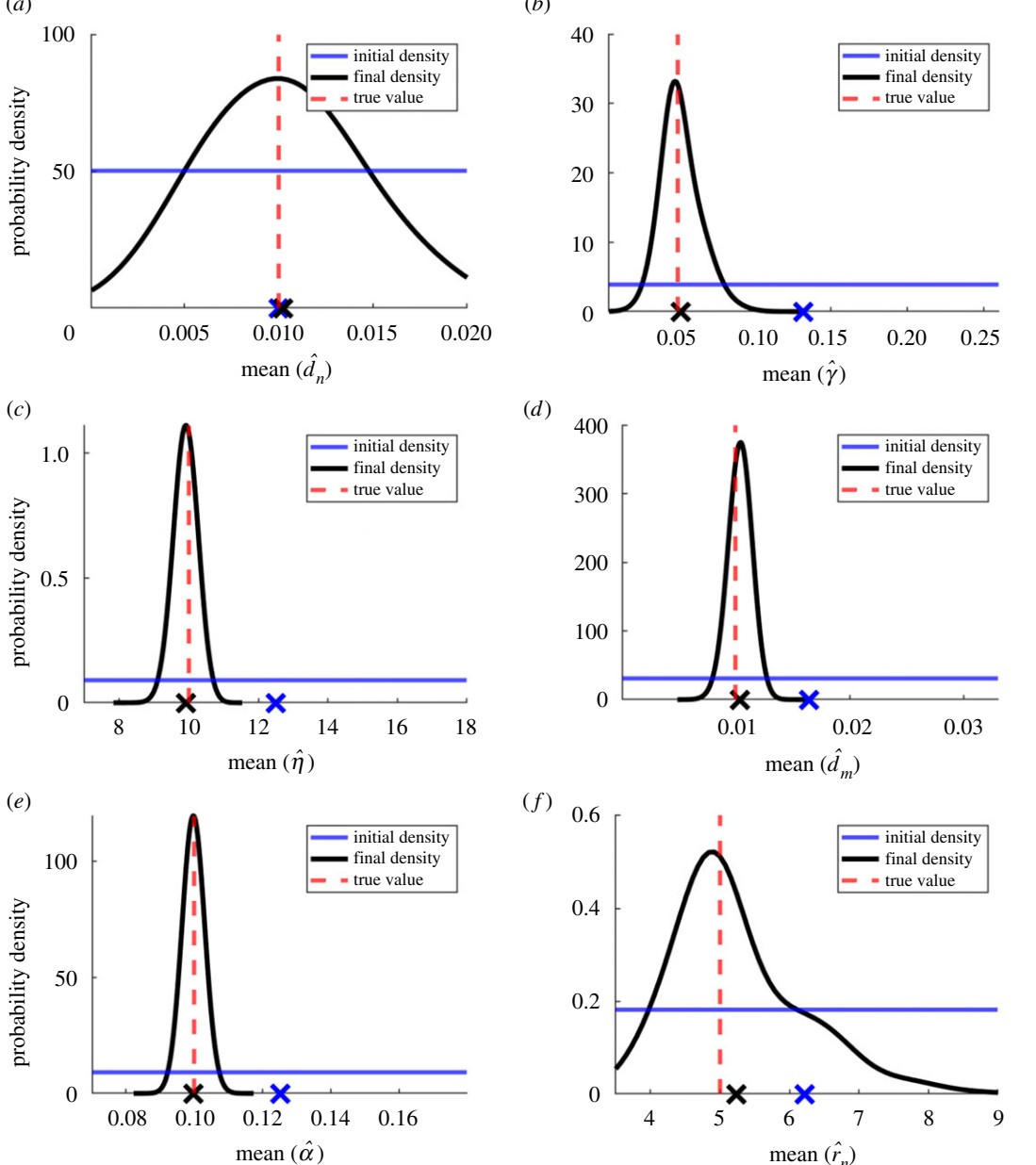

**Figure 2.** Initial (blue) and final (black) densities of the parameters in the PDE model estimated using the ABC-related scheme. Sample means are marked with 'X' on the x-axis, and the reference parameter values are shown with a red vertical dotted line. (See electronic supplementary material, appendix B (a) for the detailed density evolutions of the parameters in different rounds.)

second test, the two parameters $d_n$ and $\gamma$ associated with complex terms in the tumour cells equation were fixed; bias increased in $r_n$ and $d_m$ and slightly decreased in $\alpha$. Note that estimates of $r_n$ were poor despite it being the only parameter remaining to be estimated in the tumour cells equation. Lastly, all parameters in the tumour cells equation ($d_n$, $\gamma$ and $r_n$) were fixed; here bias in $d_m$ slightly decreased but $\alpha$ was estimated with poorer accuracy. The results in these attempts confirmed fixing certain parameters at their true values is not a solution to improve the accuracy of parameter estimates. After taking a closer look at the gradients at each evenly spaced location in the domain, averaged over time and the 200 datasets (figure 5), we noted the temporal gradients estimated by GAM shown some deviations from the true ones at certain parts of the domain, but in general, they were quite insensitive to the increase in measurement errors, except for $\partial m/\partial t$, where the deviations at the left tail seem to increase with the CV. The complex spatial gradients (e.g. second-order spatial derivatives, haptotaxis term) have also shown obvious deviations from the true ones as the CV goes up, especially at the tails of the domain. The deviated gradients can bring a negative effect to the parameter estimates, especially the ones that are associated with them. Our whole gradient matching

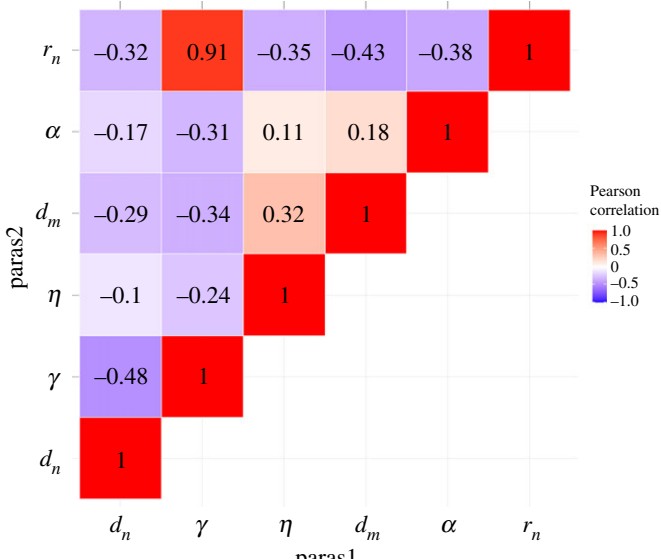

**Figure 3.** Pairwise heat maps of the final parameter densities in the first run of the ABC-related scheme on the main reference dataset.

scheme relies on the accuracy of the gradients, if the gradients obtained were inaccurate in the first place, then the parameter estimates will be poor inevitably. Constraining certain parameters when the gradients are inaccurate will make the estimates even worse since only a subset of the parameters can be altered to minimize the value of the objective function, this makes the optimization scheme lose its flexibility to a certain extent, which may affect the parameter estimates in a negative way.

## 4. Discussion

In this article, we investigated the performance of two calibration schemes for estimating the parameters of a PDE model of cancer invasion and metastasis: one related to ABC, another related to gradient matching. Both schemes were quite accurate when no perturbations were added to the reference data (table 2). In the ABC-related scheme, under multiple runs, our scheme was capable of producing consistent final results for the parameters (electronic supplementary material, table S4) showing that Monte Carlo error is low; it also produced accurate results on two datasets simulated with different parameter values (electronic supplementary material, table S5). Gradient matching produced slightly less accurate results on the reference dataset, which may be expected since the data are approximated by the smooths. Overall, both schemes gave parameter estimates at satisfactory levels; we believe if these schemes worked well with one-dimensional synthetic data, then it is certainly possible to increase the dimension and extend our case study to two dimensions, which means we can investigate more realistic invasion data (e.g. those observed in lab experiments or clinical data of cancer patients).

However, there are still some concerns at this stage. For the ABC-BCD scheme, the parameters in the tumour cells equation are estimated with the least accuracy. Possible reasons are as follows.

(i) The equation of tumour cells profile has the most complicated structure in the PDE model, increasing the difficulty of making accurate estimates of parameters within the equation.
(ii) If we look at the PDE model itself, the parameters in the tumour cells equation are all associated with numerically complex terms, it can be harder to acquire the information hidden behind the PDE model that can help to improve the accuracy of their estimates.
(iii) In this ABC-related scheme, we chose to draw parameter estimates equation by equation. As the tumour cells equation is the last one to be investigated, uncertainties produced in the previous two investigations were propagated to this final one. Therefore, although five rounds were conducted, the error rates of its parameters could still be quite high compared to others.

Another issue with the 'equation-by-equation' approach in the ABC-related scheme is that it may fail in PDE models where the equations are highly inter-dependent. In the cancer invasion model used here,

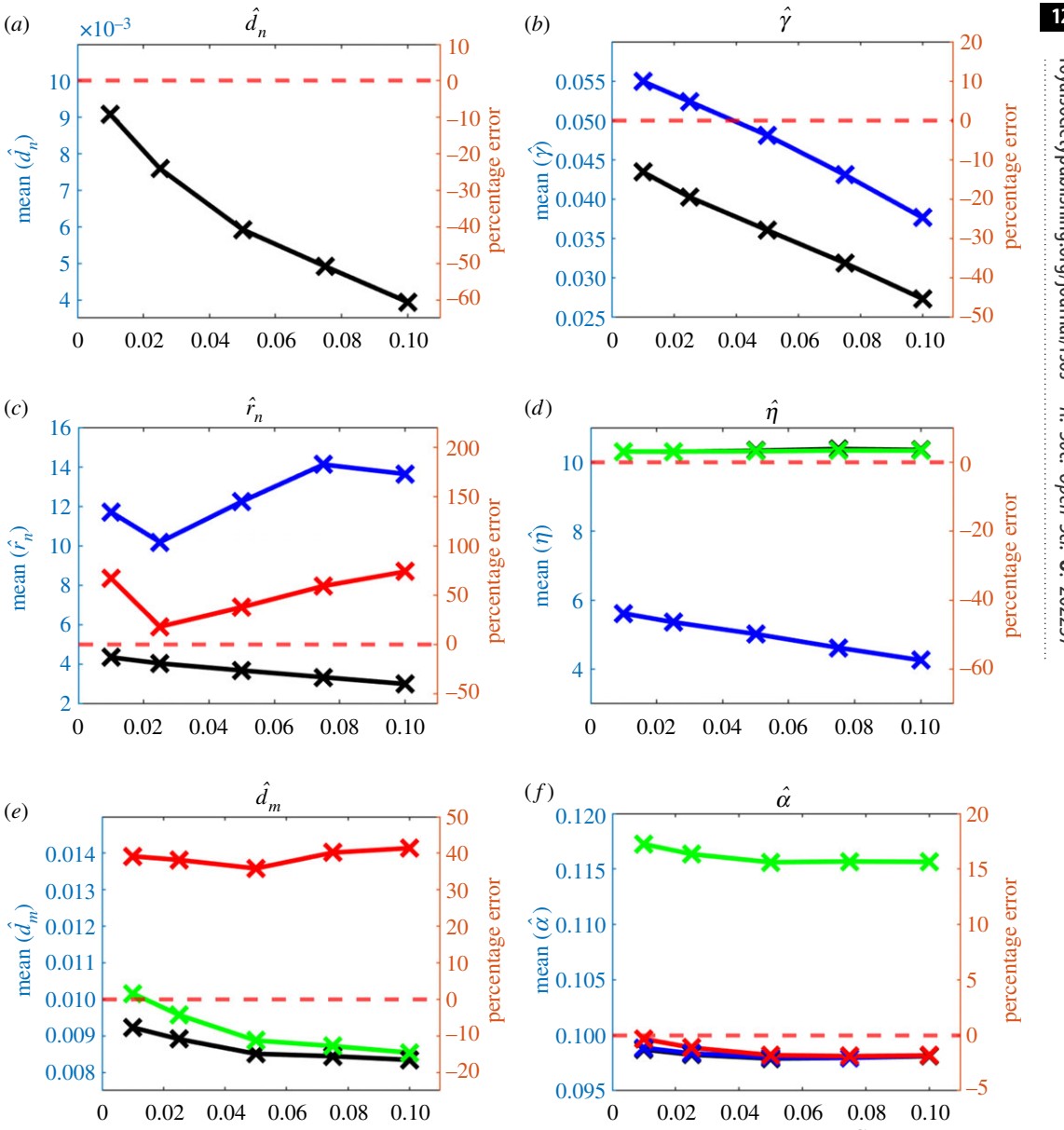

**Figure 4.** Average parameter estimates from gradient matching scheme over 200 simulated datasets at each of five levels of measurement error. Horizontal red dashed lines: true values of the parameters. Black line: parameter estimates where all parameters are estimated. Other lines show parameter estimates where some parameters are fixed at their true value: blue, $d_n$ and $d_m$ fixed; red, $d_n$ and $\gamma$ fixed; green line, $d_n$, $\gamma$ and $r_n$ fixed.

successful retrievals of reference parameter values suggests the equations are quite independent of each other. In other words, each equation in the PDE model contains the most amount of information about its own parameters.

Some other issues within our ABC scheme include the following.

(i) The reference parameter values were chosen by us, so it is quite simple to tell if our calibrations were successful via a direct comparison between the final parameter estimates and the reference values. In real-world situations, the reference parameter values will be unknown and hence the 'stopping-criterion' needs to be established more carefully and the accuracy of the final parameter estimates should be assessed with caution.

(ii) Consideration may be given to the use of a multivariate version Bhattacharyya distance or possibly other distance measuring metrics.

(iii) In the fully Bayesian setting, informative priors may be preferable under certain circumstances. When a differential equation model is involved, it can be more rigorous to propose the prior

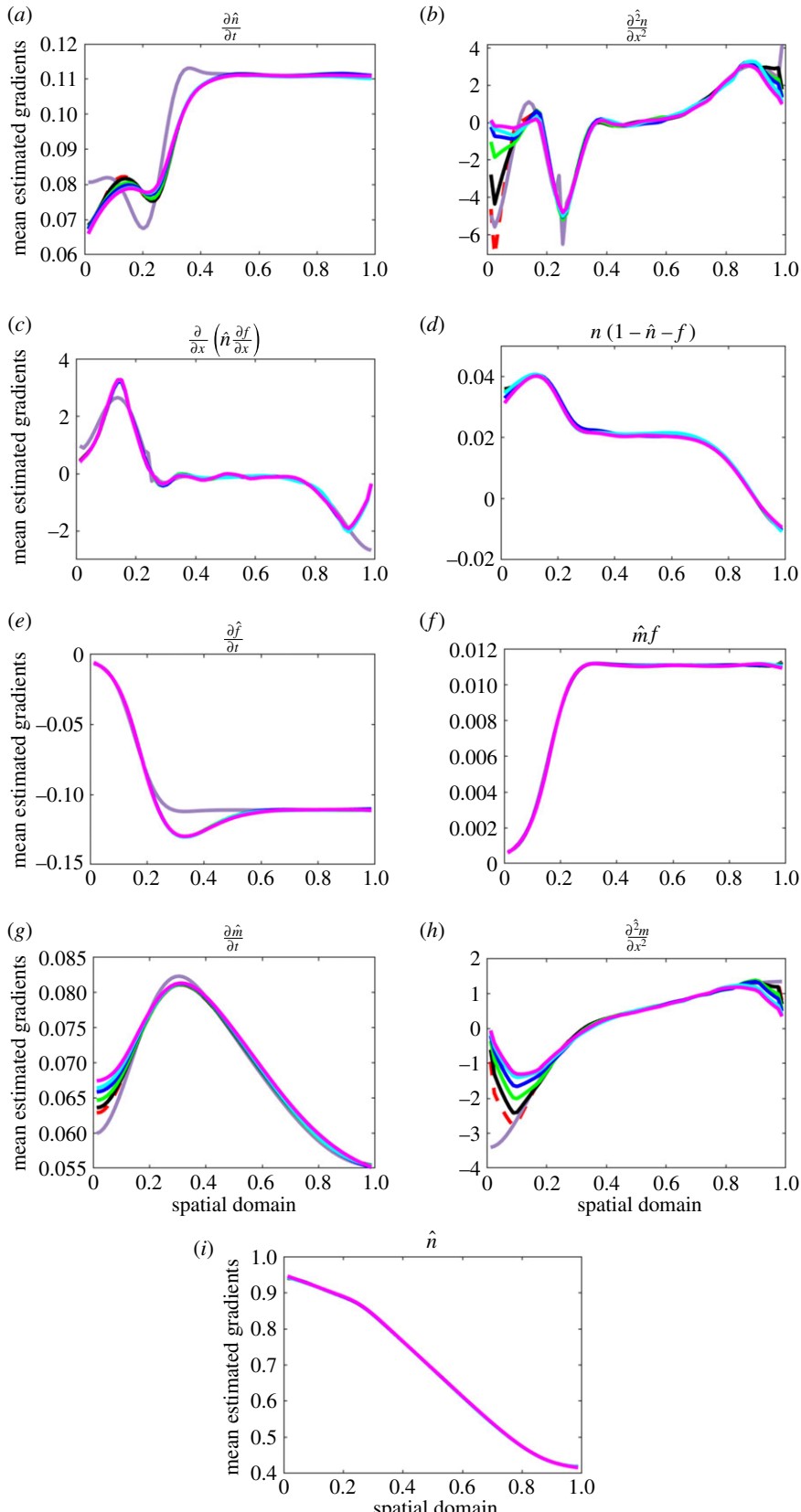

**Figure 5.** Temporal and spatial gradients in the model at different measurement error CVs. Red dashed lines: reference gradients predicted by GAM at CV equals 0. Dark purple solid lines: true gradients calculated by the finite difference scheme. Other solid lines show estimated gradients averaged over time and 200 datasets from the gradient matching scheme at CVs of 0.01 (black), 0.025 (green), 0.05 (blue), 0.075 (cyan) and 0.1 (magenta). A larger-sized version of these plots is provided in the Github repository mentioned in the 'Data accessibility' statement at the end of the manuscript.

distributions in the dimensional setting and then translate them into non-dimensional ones, since not all the priors are invariant under the change of coordinate in some cases. Fortunately, this is not a concerning issue in our work. Our setting is not completely Bayesian, the initial distributions of the parameters in this study only serve to influence the speed of convergence, they should not have an impact on the final point estimates, as long as the final point estimates are within the corresponding initial distributions.

(iv) Lastly, our current ABC scheme was not designed to measure the uncertainties in the estimates—no analytic methods exist and bootstrapping is computationally infeasible. In future work, observation errors may be introduced to the reference dataset so uncertainties in parameter estimates can be assessed using a full posterior inference under the Bayesian framework. However, we realize the incorporation of observation errors can be a challenging problem. Alahmadi *et al.* [45] have argued the posterior distributions derived from those ABC methods which fail to adequately model the measurement errors may not accurately reflect the epistemic uncertainties in parameter values. Hence, it is necessary to model and incorporate the errors in the correct way when using ABC methods.

The gradient matching scheme has the same issue as the ABC-related scheme when estimating numerically complex terms. Our attempts to fix some parameters at their true values and estimate only a subset largely did not produce a substantial increase in accuracy of the estimated parameters. The gradients averaged over time and the 200 datasets gave us a hint of what might have gone wrong, hence we investigated the gradients in a more explicit way (see electronic supplementary material, figures S10–S18). In the tumour cells-related gradients, the temporal ones estimated from the fitted GAM (electronic supplementary material, figure S10) are mostly consistent across the domain at the later time points, but some obvious deviations can be observed at the left tail of the domain at the early time points. The deviation between the true gradients calculated by the finite difference scheme and the gradients estimated from the fitted GAM with no measurement error added was quite obvious. This difference between the GAM-estimated and true gradients is responsible for the errors in the tumour cell-related estimates when no perturbation was added. The same thing happens for the spatial gradients of $\gamma$ (electronic supplementary material, figure S12): the gradients predicted by GAM under different CVs were quite consistent, but some obvious difference between the GAM gradients and the true gradients can be observed at early and middle time points. The second-order spatial gradients related to $d_n$ (electronic supplementary material, figure S11) were mostly deviated from the true gradients as the CV goes up, especially at the early time points. The logistic growth gradients related to $r_n$ (electronic supplementary material, figure S13) are the only robust ones among all the tumour cells gradients, yet one can still observe some minor deviations at the left tail of the domain in the early time points. In summary, a considerable proportion of the tumour cells-related gradients were estimated incorrectly by GAM, especially at the early time points and the tails in the domain. This bias increased as the level of measurement errors went up, which may explain the rapid fall in accuracy of the related parameters as the CV goes up. On the other hand, the ECM-related gradients (electronic supplementary material, figures S14 and S15) were consistently estimated by GAM at all time points under all different levels of CVs; this coincides with the $\eta$ estimates which are quite insensitive to the change in CV. The difference between the temporal gradients estimated by GAM and the ones calculated by the finite difference scheme at the early time points may explain the errors occurred in the gradient matching estimates when no perturbation was added. Lastly, in the MDE-related gradients, the temporal ones (electronic supplementary material, figure S16) estimated by GAM are mostly consistent at the middle time points, although some deviations in the left tail of the domain can be observed at $t = 1$. At $t = 9$, the gradients deviated from the reference ones but following the same shape. The spatial gradients of $d_m$ are mostly consistent in the middle part of the domain across all the time points; however, quite obvious deviations can be observed in the tails of the domain, especially when the CV is high. Examining the difference between the gradients estimated by GAM and the ones calculated by the finite difference scheme at different time points, we may tell the gradients at both tails were over-smoothed, this may be the reason of the poor accuracy in the $d_m$ estimate when no perturbations were added. On the other hand, comparing the temporal and second-order spatial gradients estimated by GAM in the tumour cells equation, the same gradients in the MDE equation were much better estimated; this can explain the estimates of $d_m$ being more accurate and stable than $d_n$ as the CV increases. The spatial gradients of $\alpha$ are mostly consistent, which explains the consistent estimates of $\alpha$ in response to the increasing level of CV.

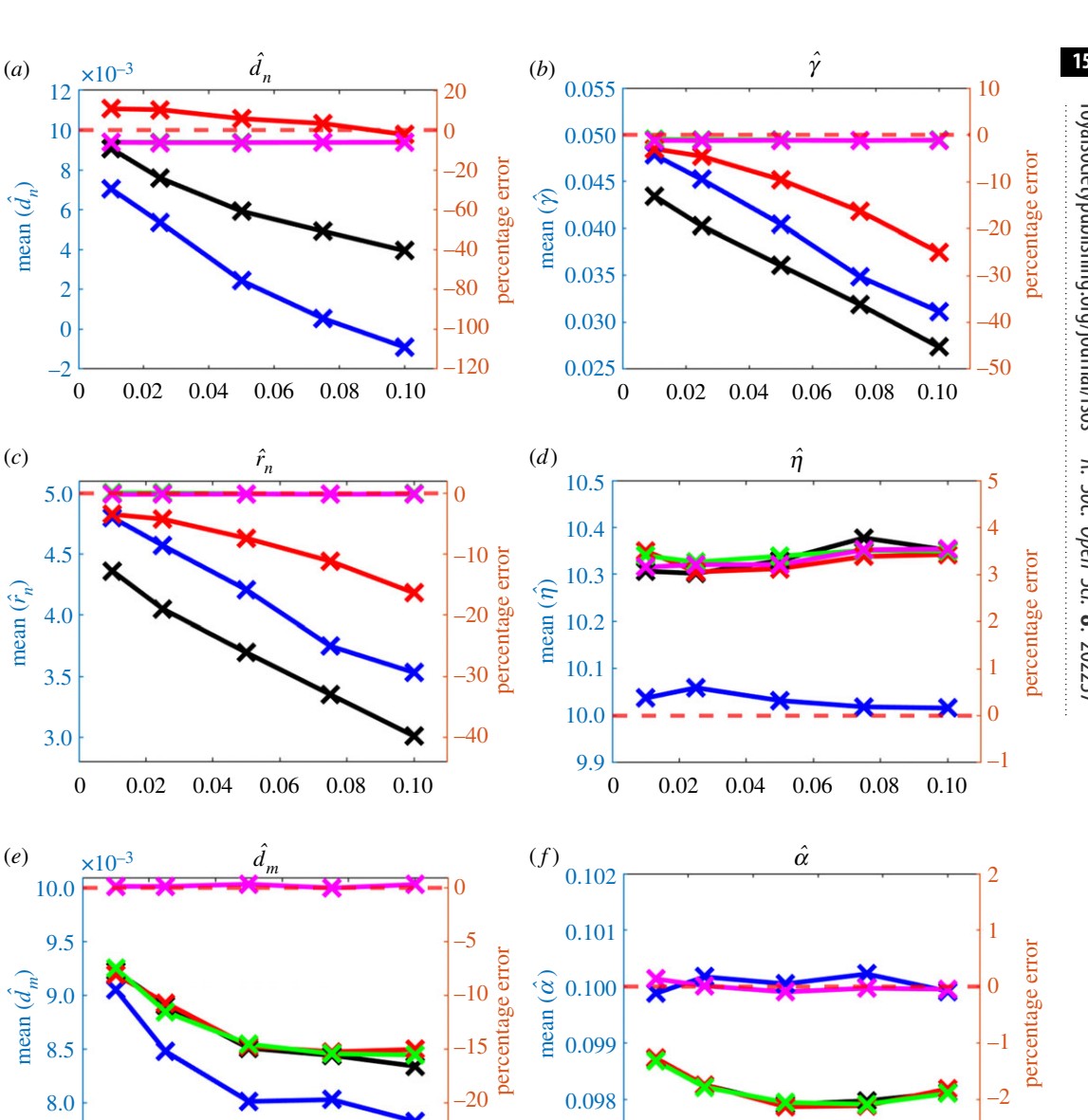

**Figure 6.** Average parameter estimates from gradient matching scheme over 200 simulated datasets at each of five levels of measurement error. Black line: parameter estimates where all parameters are estimated with the gradients fitted by GAM. Other lines show parameter estimates where some gradients are replaced by the reference ones or truncated: blue, truncated gradients; red, tumour cells temporal gradients, $d_n$ and $\gamma$ spatial gradients replaced by the reference ones; green, all tumour cells-related gradients replaced by the reference ones; magenta, all tumour cells-related gradients and $d_m$ spatial gradients replaced by the reference ones.

Instead of fixing the parameters at their true values, we tried two possible methods to improve the parameter estimates:

(i) Cut off the parts with obvious deviated gradients ($t = 1, 2, 8, 9$, the first and last 20 locations in the domain; the gradient matrix is now truncated from $9 \times 78$ to $5 \times 38$).

(ii) Fix certain gradients at their true values. The results we obtained with this approach were much better than the ones seen in the sensitivity test (figure 6).

In general, replacing the inaccurate gradients by the true gradients calculated by a finite difference scheme is a much better option than truncating the gradient matrices, as it turned out only the estimates of $\eta$ became significantly better with the later option, possibly due to the simple structure

and the consistent estimates of ECM-related gradients in the middle time points and the middle part of the domain. In the very last attempt of gradient replacements, where all the tumour cells-related gradients, the temporal gradients of MDE and the second-order spatial gradients of $d_m$ are replaced by the reference ones, the accuracy of tumour cells-related parameters improved notably and became much more stable, the other parameters maintained at similar levels of accuracy and are quite insensitive to the increase of CV. Hence, in order to improve the accuracy of all parameters, we may need a more sophisticated fitting method which can estimate the tumour cells-related gradients better.

A common concern for the two schemes proposed in the current work is that the synthetic data were generated from a particular model and then fitted back to the same model. In a real-world case of real cancer data, our models are simply approximations, and we will want to consider the model selection problem as well as goodness-of-fit of the model to the data. We also side-stepped the important issue of uncertainty quantification. A real-world application will at least wish to consider quantifying uncertainty arising from measurement errors (e.g. [45]) and may also attempt to quantify other sources of uncertainty such as model mis-specification error.

Overall, we believe the parameter estimates for PDE models using statistical approaches is a strong alternative to searching high-dimensional parameter space manually, which requires a very powerful numerical scheme if the differential equations system is complicated. The optimization schemes presented in this work pave the way for applying similar schemes to real cancer invasion and metastastic spread data (e.g. from *in vitro* organotypic assays) [46]. Also, we believe the power of our optimization schemes is not limited to this specific model of cancer invasion; it can certainly be applied to other PDE models. Although the ideas behind the methods proposed here work, the approach may still be considered to be at an early stage, and there are many questions remaining to be answered.

Data accessibility. Data and relevant code for this research work are stored in Github: https://github.com/ycx12341/Data-Code-Figures-ver-4 and have been archived within the Zenodo repository: https://doi.org/10.5281/zenodo.4640385.

Authors' contributions. Y.X. developed the calibration schemes and ran the analysis, with substantive input from L.T. and M.A.J.C. All three authors drafted the paper.

Competing interests. At the time of writing, Prof. Mark Chaplain and Prof. Len Thomas are Board Members of Royal Society Open Science, but had no involvement in the review or assessment of the paper.

Funding. Y.X. is funded by a Doctoral Training Partnership grant from the Engineering and Physical Sciences Research Council (EPSRC) and a University of St Andrews St Leonard's International Fee Scholarship.

Acknowledgements. We thank Richard Glennie and Dave Campbell for valuable discussions, and two anonymous reviewers for their helpful comments and suggestions. Y.X. thanks Yuxuan Wang for his great help with Github.

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
