## [Peer Review File · Royal Society Open Science]

Review History

RSOS-202237.R0 (Original submission)

Review form: Reviewer 1

Is the manuscript scientifically sound in its present form?

Yes

Are the interpretations and conclusions justified by the results?

Yes

Is the language acceptable?

Yes

Do you have any ethical concerns with this paper?

No

Have you any concerns about statistical analyses in this paper?

No

Recommendation?

Major revision is needed (please make suggestions in comments)

Comments to the Author(s)

Please see the attached file (Appendix A).

Review form: Reviewer 2**Is the manuscript scientifically sound in its present form?**

No

Are the interpretations and conclusions justified by the results?

Yes

Is the language acceptable?

Yes

Do you have any ethical concerns with this paper?

No

Have you any concerns about statistical analyses in this paper?

Yes

Recommendation?

Major revision is needed (please make suggestions in comments)

Comments to the Author(s)

This paper investigates statistical inference for a PDE cancer model using approximate Bayesian computation (ABC) and gradient matching. It provides an interesting comparison of these methods on a challenging application.

I have one major concern about this paper: the correctness of the ABC algorithm used. Therefore I recommend major corrections. Otherwise the paper performs a good simulation study and is well written, so I have only a few other minor comments. Details of major and minor comments are below.

Also if the authors have time in future, it would be interesting to extend this study, especially to investigate how ABC performs in the case of observation error. Furthermore I wonder if ABC is the best algorithm to use for optimization when the simulator is deterministic (i.e. when there is no observation error). For instance one could use stochastic approximation optimization methods (of the Kiefer-Wolfowitz type). However exploration of these issues is beyond the scope of what's needed for this paper to be accepted.

Major comment

=====

The ABC algorithm used has some non-standard features compared to standard ABC-SMC algorithm (for example, see the review article of Marin et al 2012 <https://doi.org/10.1007/s11222-011-9288-2>). One of these features seems like it could invalidate

the correctness of the algorithm for this application (see first point below). The others seem problematic for general use of ABC, but not this particular application (see second point below).

ABC bandwidth

The paper's ABC algorithm uses $\text{weight} = \text{distance}^{-1/2}$. ABC-SMC algorithms would typically include a bandwidth variable h here e.g. $\text{weight} = \text{distance}^{-1/2h}$ which is reduced as the iterations of the algorithm progress (early papers used a pre-specified schedule of values, but reducing h adaptively is now more common). This is necessary for the algorithm to converge on the true parameter values.

I suggest using a reducing bandwidth here and rerunning the simulations. Alternatively the authors could explain why they wish to use a fixed bandwidth, and adjust this discussion accordingly i.e. note that the algorithm does not converge on the true parameters, but instead gives approximate results, and justify why this is good enough for the current purposes. However in the latter case, I think the issues in the next point become more important.

Other issues with algorithm

Some other unusual features of the ABC algorithm are:

- * The weights do not include a term for the density of the proposed parameters.
- * Weights are not rescaled by the standard formula (dividing by the sum of the weights).

This means that the usual sequential Monte Carlo / population Monte Carlo analysis of the algorithm does not hold and so this algorithm does not output samples from the usual ABC approximation to the posterior density. It is somewhat unclear therefore what distribution the algorithm samples from. In general it would be preferable to have some reassurance that this algorithm does not have some undesirable properties compared to standard ABC e.g. large extra approximation error, or no guarantee of convergence.

However for their particular application, the authors simply wish to use ABC as an optimisation algorithm. Thus, so long as it finds parameter values which approximately minimise the distance, then this is sufficient for the desired behaviour. (However to do so, the bandwidth issue above probably needs to be addressed.)

Minor comments

=====

1. It would be preferable to show numerical results using mathematical notation e.g. 6.9×10^6 rather than $6.9e6$. This is particularly useful for numbers which can simply be represented as decimals e.g. 0.012 is much easier to interpret in a table than $1.2e-2$.
2. It might be worth briefly discussing the recent paper "A comparison of approximate versus exact techniques for Bayesian parameter inference in nonlinear ordinary differential equation models" by Alahmadi et al (<https://doi.org/10.1098/rsos.191315>). They discuss how ABC methods can fail for differential equation models if badly implemented. Although this problem applies mainly to the case of observation error, it might be interesting to point this out to readers of your paper.

3. The authors say in a few places that ABC is not suitable for assessing uncertainty in the parameter estimates. Could the authors expand on what type of uncertainty they wish to capture here? From a Bayesian or maximum likelihood viewpoint, if the likelihood is a spike at a single parameter value then there is no uncertainty! Other sources of uncertainty exist, such as model misspecification, but are quite hard to capture by any method

4. Page 2 says "Under the assumption that the differential equation model is correct and that the observations arise independently from a normal distribution then the least squares parameter estimates are maximum likelihood estimates." I think it is not necessary for the differential equation model to be correct for the least squares estimates and MLE to coincide.

5. Page 6 says "the algorithm will eventually converge on the best fitting values with no variation between parameter sets". As well as the bandwidth issue mentioned above, another issue with this is the use of summary statistics. There's an implicit assumption here that an exact match of summary statistics implies an exact match of the full data. This should be mentioned, with a brief discussion of how reasonable this is for the application under investigation.

6. Page 9. Mention which optimization algorithm was used when using the optim function in R.

7. Page 11 says "Constraining certain parameters when the gradients are inaccurate will make the estimates even worse, since this makes the optimization scheme loses its ability to adjust itself." Can you give some more explanation or reword here? It's not obvious to me that this is true or exactly what "the ability to adjust itself" means.

8. Page 12 says "If we look at the PDE model itself, the parameters in the tumour cells equation are all associated with numerically complex terms, and thus the information hidden behind the PDE model that can help to improve the accuracy of their estimates is therefore limited." Reword this? Currently it seems to say that numerical complexity implies lack of informativeness, which seems incorrect.

9. In Figure 4, on page 13, it's hard to make out different lines. This is better in the larger sized versions of these plots at the end of the document. Maybe mention that these are available in the caption to Figure 4? Also, the ^ character seems to be misplaced in several of the plot headings.

10. Page 13 says "However, this difference here should be accounted for the errors in the tumour cell-related estimates when no perturbation was added." I didn't understand what this sentence meant. Is it possible to explain further, or is there a typo in the sentence?

11. Page 15 says "Overall, we believe the parameter estimates for PDE models using statistical approaches is a strong alternative to searching high-dimensional parameter space by hand-tuning". What is meant by "hand-tuning" here? I couldn't find this phrase anywhere earlier in the paper.

Possible typos

=====

1. Page 6: "suitable for estimating parameter values but ****not not**** for assessing uncertainty on the estimates"

2. Page 8: "The discrepancies between the two sides of the equations ****was**** then calculated"

3. Page 10: "but in general, they were quite insensitive to the increase in measurement errors, except for $\partial m / \partial t$, the deviations at the left tail seem to increase with the CV." Is there a word missing somewhere here e.g. should it be "...****where**** the deviations..."?

4. Page 11: "The complex spatial gradients (e.g. second order spatial derivatives, haptotaxis term) also **shown** obvious deviations"

Decision letter (RSOS-202237.R0)

Dear Mr Xiao

The Editors assigned to your paper RSOS-202237 "Calibrating models of cancer invasion: parameter inference using Approximate Bayesian Computation and gradient matching" have now received comments from reviewers and would like you to revise the paper in accordance with the reviewer comments and any comments from the Editors. Please note this decision does not guarantee eventual acceptance.

Please submit your revised manuscript and required files (see below) no later than 21 days from today's (ie 02-Mar-2021) date. Note: the ScholarOne system will 'lock' if submission of the revision is attempted 21 or more days after the deadline. If you do not think you will be able to meet this deadline please contact the editorial office immediately.

on behalf of Professor Tim Rogers (Associate Editor)
openscience@royalsociety.org

Reviewer comments to Author:

Reviewer: 1

Comments to the Author(s)

Please see the attached file.

Reviewer: 2

Comments to the Author(s)

This paper investigates statistical inference for a PDE cancer model using approximate Bayesian computation (ABC) and gradient matching. It provides an interesting comparison of these methods on a challenging application.

I have one major concern about this paper: the correctness of the ABC algorithm used. Therefore I recommend major corrections. Otherwise the paper performs a good simulation study and is well written, so I have only a few other minor comments. Details of major and minor comments are below.

Also if the authors have time in future, it would be interesting to extend this study, especially to investigate how ABC performs in the case of observation error. Furthermore I wonder if ABC is the best algorithm to use for optimization when the simulator is deterministic (i.e. when there is no observation error). For instance one could use stochastic approximation optimization methods (of the Kiefer-Wolfowitz type). However exploration of these issues is beyond the scope of what's needed for this paper to be accepted.

Major comment

=====

The ABC algorithm used has some non-standard features compared to standard ABC-SMC algorithm (for example, see the review article of Marin et al 2012 <https://doi.org/10.1007/s11222-011-9288-2>). One of these features seems like it could invalidate the correctness of the algorithm for this application (see first point below). The others seem problematic for general use of ABC, but not this particular application (see second point below).

ABC bandwidth

The paper's ABC algorithm uses $\text{weight} = \text{distance}^{-1/2}$. ABC-SMC algorithms would typically include a bandwidth variable h here e.g. $\text{weight} = \text{distance}^{-1/2h}$ which is reduced as the iterations of the algorithm progress (early papers used a pre-specified schedule of values, but reducing h adaptively is now more common). This is necessary for the algorithm to converge on the true parameter values.

I suggest using a reducing bandwidth here and rerunning the simulations. Alternatively the authors could explain why they wish to use a fixed bandwidth, and adjust this discussion accordingly i.e. note that the algorithm does not converge on the true parameters, but instead gives approximate results, and justify why this is good enough for the current purposes. However in the latter case, I think the issues in the next point become more important.

Other issues with algorithm

Some other unusual features of the ABC algorithm are:

- * The weights do not include a term for the density of the proposed parameters.
- * Weights are not rescaled by the standard formula (dividing by the sum of the weights).

This means that the usual sequential Monte Carlo / population Monte Carlo analysis of the algorithm does not hold and so this algorithm does not output samples from the usual ABC approximation to the posterior density. It is somewhat unclear therefore what distribution the algorithm samples from. In general it would be preferable to have some reassurance that this algorithm does not have some undesirable properties compared to standard ABC e.g. large extra approximation error, or no guarantee of convergence.

However for their particular application, the authors simply wish to use ABC as an optimisation algorithm. Thus, so long as it finds parameter values which approximately minimise the distance, then this is sufficient for the desired behaviour. (However to do so, the bandwidth issue above probably needs to be addressed.)

Minor comments

=====

1. It would be preferable to show numerical results using mathematical notation e.g. 6.9×10^6 rather than $6.9e6$. This is particularly useful for numbers which can simply be represented as decimals e.g. 0.012 is much easier to interpret in a table than $1.2e-2$.
2. It might be worth briefly discussing the recent paper "A comparison of approximate versus exact techniques for Bayesian parameter inference in nonlinear ordinary differential equation models" by Alahmadi et al (<https://doi.org/10.1098/rsos.191315>). They discuss how ABC methods can fail for differential equation models if badly implemented. Although this problem applies mainly to the case of observation error, it might be interesting to point this out to readers of your paper.
3. The authors say in a few places that ABC is not suitable for assessing uncertainty in the parameter estimates. Could the authors expand on what type of uncertainty they wish to capture here? From a Bayesian or maximum likelihood viewpoint, if the likelihood is a spike at a single parameter value then there is no uncertainty! Other sources of uncertainty exist, such as model misspecification, but are quite hard to capture by any method
4. Page 2 says "Under the assumption that the differential equation model is correct and that the observations arise independently from a normal distribution then the least squares parameter estimates are maximum likelihood estimates." I think it is not necessary for the differential equation model to be correct for the least squares estimates and MLE to coincide.
5. Page 6 says "the algorithm will eventually converge on the best fitting values with no variation between parameter sets". As well as the bandwidth issue mentioned above, another issue with this is the use of summary statistics. There's an implicit assumption here that an exact match of summary statistics implies an exact match of the full data. This should be mentioned, with a brief discussion of how reasonable this is for the application under investigation.
6. Page 9. Mention which optimization algorithm was used when using the optim function in R.
7. Page 11 says "Constraining certain parameters when the gradients are inaccurate will make the estimates even worse, since this makes the optimization scheme loses its ability to adjust itself."

Can you give some more explanation or reword here? It's not obvious to me that this is true or exactly what "the ability to adjust itself" means.

8. Page 12 says "If we look at the PDE model itself, the parameters in the tumour cells equation are all associated with numerically complex terms, and thus the information hidden behind the PDE model that can help to improve the accuracy of their estimates is therefore limited." Reword this? Currently it seems to say that numerical complexity implies lack of informativeness, which seems incorrect.

9. In Figure 4, on page 13, it's hard to make out different lines. This is better in the larger sized versions of these plots at the end of the document. Maybe mention that these are available in the caption to Figure 4? Also, the ^ character seems to be misplaced in several of the plot headings.

10. Page 13 says "However, this difference here should be accounted for the errors in the tumour cell-related estimates when no perturbation was added." I didn't understand what this sentence meant. Is it possible to explain further, or is there a typo in the sentence?

11. Page 15 says "Overall, we believe the parameter estimates for PDE models using statistical approaches is a strong alternative to searching high-dimensional parameter space by hand-tuning". What is meant by "hand-tuning" here? I couldn't find this phrase anywhere earlier in the paper.

Possible typos

=====

1. Page 6: "suitable for estimating parameter values but **not not** for assessing uncertainty on the estimates"

2. Page 8: "The discrepancies between the two sides of the equations **was** then calculated"

3. Page 10: "but in general, they were quite insensitive to the increase in measurement errors, except for $\partial m/\partial t$, the deviations at the left tail seem to increase with the CV." Is there a word missing somewhere here e.g. should it be "...**where** the deviations...?"

4. Page 11: "The complex spatial gradients (e.g. second order spatial derivatives, haptotaxis term) also **shown** obvious deviations"

===PREPARING YOUR MANUSCRIPT===

While not essential, it will speed up the preparation of your manuscript proof if accepted if you format your references/bibliography in Vancouver style (please see

<https://royalsociety.org/journals/authors/author-guidelines/#formatting>). You should include DOIs for as many of the references as possible.

===PREPARING YOUR REVISION IN SCHOLARONE===

Author's Response to Decision Letter for (RSOS-202237.R0)

See Appendix B.

RSOS-202237.R1 (Revision)

Review form: Reviewer 1

Is the manuscript scientifically sound in its present form?

Yes

Are the interpretations and conclusions justified by the results?

Yes

Is the language acceptable?

Yes

Do you have any ethical concerns with this paper?

No

Have you any concerns about statistical analyses in this paper?

No

Recommendation?

Accept as is

Comments to the Author(s)

The authors have substantially addressed every concern that I raised in my initial report. Thus, I am of the view that this manuscript is now suitable for publication in Royal Society Open Science.

Review form: Reviewer 2

Is the manuscript scientifically sound in its present form?

Yes

Are the interpretations and conclusions justified by the results?

Yes

Is the language acceptable?

Yes

Do you have any ethical concerns with this paper?

No

Have you any concerns about statistical analyses in this paper?

No

Recommendation?

Accept with minor revision (please list in comments)

Comments to the Author(s)

The authors have addressed all the points from my review and I am happy to recommend that the paper is accepted for publication.

I have two minor comments which the authors could perhaps incorporate in their final draft. These are listed below using the numbering scheme from the authors' response.

R2.2 It might be worth mentioning that there are some differences between your algorithm and standard ABC-SMC, so that readers are aware that some modifications are needed if they are interested in inferring the full posterior.

R2.13 This sentence is clearer now: "However, the difference between the GAM predicted and true gradients should be accounted for the errors in the tumour cell-related estimates when no perturbation was added." But I am still unsure of what is meant by "should be accounted for". Does this mean it should be adjusted somehow?

Decision letter (RSOS-202237.R1)

Dear Mr Xiao

On behalf of the Editors, we are pleased to inform you that your Manuscript RSOS-202237.R1 "Calibrating models of cancer invasion: parameter estimation using Approximate Bayesian Computation and gradient matching" has been accepted for publication in Royal Society Open Science subject to minor revision in accordance with the referees' reports. Please find the referees' comments along with any feedback from the Editors below my signature.

Please submit your revised manuscript and required files (see below) no later than 7 days from today's (ie 18-May-2021) date. Note: the ScholarOne system will 'lock' if submission of the revision is attempted 7 or more days after the deadline. If you do not think you will be able to meet this deadline please contact the editorial office immediately.

on behalf of Professor Tim Rogers (Associate Editor)
openscience@royalsociety.org

Associate Editor Comments to Author (Professor Tim Rogers):

Associate Editor: 1

Comments to the Author:

The final revisions suggested by the referee are optional.

Reviewer comments to Author:

Reviewer: 2

Comments to the Author(s)

The authors have addressed all the points from my review and I am happy to recommend that the paper is accepted for publication.

I have two minor comments which the authors could perhaps incorporate in their final draft. These are listed below using the numbering scheme from the authors' response.

R2.2 It might be worth mentioning that there are some differences between your algorithm and standard ABC-SMC, so that readers are aware that some modifications are needed if they are interested in inferring the full posterior.

R2.13 This sentence is clearer now: "However, the difference between the GAM predicted and true gradients should be accounted for the errors in the tumour cell-related estimates when no perturbation was added." But I am still unsure of what is meant by "should be accounted for". Does this mean it should be adjusted somehow?

Reviewer: 1

Comments to the Author(s)

The authors have substantially addressed every concern that I raised in my initial report. Thus, I am of the view that this manuscript is now suitable for publication in Royal Society Open Science.

===PREPARING YOUR MANUSCRIPT===

===PREPARING YOUR REVISION IN SCHOLARONE===

Author's Response to Decision Letter for (RSOS-202237.R1)

See Appendix C.

Decision letter (RSOS-202237.R2)

Dear Mr Xiao,

I am pleased to inform you that your manuscript entitled "Calibrating models of cancer invasion: parameter estimation using Approximate Bayesian Computation and gradient matching" is now accepted for publication in Royal Society Open Science.

on behalf of Professor Tim Rogers (Associate Editor)
openscience@royalsociety.org

Appendix A

A review of “Calibrating models of cancer invasion: parameter inference using Approximate Bayesian Computation and gradient matching” by Y Xio, L Thomas and MAJ Chaplain.

This is an interesting and exciting piece of Mathematical Biology that has been a pleasure to read. The authors review the construction of a Partial Differential Equation (PDE) model of cancer invasion, and they then proceed to fit parameters to this model. Parameter fitting is carried out with both an Approximate Bayesian Computation (ABC) and a gradient matching technique. The thrust of this manuscript is in demonstrating a successful parameter inference pipeline. This approach is likely to be emulated by other authors.

There are clear strengths to this expository work, and these are set out, section-by-section, in the following paragraphs. As this work introduces well-described ABC inference techniques to the Mathematical Biology literature, a number of directions for future work are suggested for the author’s consideration. Whilst this reviewer has considered the manuscript in its entirety, this feedback focuses on the ABC inference component of the work.

Introduction. The authors start by explaining that “a common issue” that arises when using systems of differential equations to model various phenomena is that “some or all model parameters are not known”. They then say that model fitting techniques can provide the missing parameters. On a first reading, it appears that only point parameter estimates are required for the Mathematical Biology case study that is to follow. For many modelling applications, such as the study eloquently set out in the next section, this is perfectly sensible. Then, the authors raise the ambition of operating in a fully Bayesian setting -- setting priors on the parameters, observing experimental data, and then arriving at a posterior distribution to fully characterise the parameters. The authors might wish to explore the possibility of using the entire posterior distribution to mark the uncertainty in the parameters, for example, by considering the posterior predictive distribution. The authors might wish to highlight the circumstances where a fully Bayesian approach would be advantageous.

The authors might wish to make a small number of technical changes to this section. For example, the authors ought to formalise their references to “the best parameter values”, perhaps by explaining that they make use of Bayes estimators. Further details, such as the appropriate loss function the authors have considered, could be specified to ensure a complete description of their work.

Methods. The PDE system is set out with stunning clarity and a standard non-dimensionalisation is carried out. The description of the ABC method could, perhaps, be edited to make full use of the opportunities offered by Bayesian inference. Specifically:

- (a) Step (i) imposes a uniform prior on the parameters in the non-dimensional setting. Whilst a uniform prior is frequently encountered in this line of research, there are circumstances where alternative priors are better-suited. For example, there are occasions where it is necessary to use an informative prior.

The authors might wish to acknowledge that there are specific circumstances where it is necessary to first impose priors in the dimensional PDE setting, and then to translate these priors into the non-dimensional PDE formulation. This is a consequence of not all priors being invariant under a change of coordinates.

- (b) The observant reader might recognise the similarities between the author’s approach, and the ABC Sequential Monte Carlo (ABC-SMC) method contained in reference [15]. The authors might wish to include a few sentences discussing the similarities and differences.

- (c) The authors do not seek the entire posterior distribution, but rather, they zoom in on a mean value. This has the advantage of providing a point estimate parameter for the model. For certain models this would mean the opportunity to fully characterise the uncertainty in the parameter-set is passed up, and a valid posterior density is not provided. This arises as the parameters are sampled according to a resampling probability (see part (iii)(a)(iv)) rather than the prior distribution, and presumably not re-weighted accordingly. The authors are up-front about this: for example, in Figure 2, an “initial density” and a “final density” are shown.
- (d) The authors have computed resampling weights by first determining $p^{-0.5}$, and then linearly re-scaling. It is clear that the authors arrived at this choice after exhaustive investigations. To ensure that full credit is given to their findings, the authors might wish to provide a brief motivation for this choice (especially of the exponent shown above). It might also be worth mentioning that K can be j -dependent.
- (e) The authors are right to acknowledge that a “stopping rule” has not been specified. A very simple option would be to plot the mean parameter as the algorithm unfolds, and to use this as a basis for a simple stopping rule. It is not necessary to address this in full in this manuscript, and a brief description of how a stopping rule could be implemented would probably satisfy most readers.

The reviewer has no specific comments concerning the gradient matching scheme set out in this section.

Results. This section carefully describes the outputs of the authors’ investigations. In short, the authors picked specific parameters, generated *in silico* data, and then sought to recover the parameters with the inference algorithm. To ensure that readers fully appreciate the benefits of ABC inference, the authors might wish to consider:

- (a) The difference between the picked and recovered parameters is referred to as a “percentage error”. This is a snap-shot of the algorithm’s performance at a specified parameter-set, and, in this case, in the absence of any measurement error. The reviewer would like to raise the possibility of considering the mean squared error of the outputs, so that the error can be naturally decomposed into bias and variance components.
- (b) Only the marginal “final densities” are shown. An interested reader might wish to see pairwise heatmaps of “final densities” too.
- (c) It is unfortunate that, due to computational constraints, it has not been possible to demonstrate the performance of the ABC method on a noisy dataset. Building the computer code to carry out this research must have been a formidable task, and the reviewer is satisfied that every effort would have been made to ensure its efficiency.

Discussion. This final section is refreshingly realistic in its description of the manuscript’s findings. The authors are right to raise “the model selection problem” as the next frontier in mechanistic modelling. Whilst it would be unfair to ask the authors to detail their future research plans, it might be helpful if, by way of example, they are proposing to make use of Bayes factors or similar tools to conduct this analysis.

Appendix B

Dear Professor Rogers,

Thank you for your initial decision on our manuscript, and for the very helpful comments and suggestions provided by the two reviewers. We attach a revised version that addresses all the points made. Below we reproduce each comment (in regular font) and how we dealt with it (in *red italic font, with underlining used to indicate text changes in the manuscript*). We have numbered the comments (R1.1, R1.2, ... for reviewer 1 and R2.1, R2.2, ... for reviewer 2) to make them easier to track. We look forward to hearing your thoughts on our revised submission.

With best wishes,

Yunchen Xiao and coauthors

Comments	Status	Actions
R1.1	✓	No actions required
R1.2	✓	Text revised as suggested
R1.3	✓	Text revised as suggested
R1.4	✓	Text revised as suggested
R1.5	✓	Additional text provided as suggested
R1.6	✓	Algorithm revised as suggested
R1.7	✓	Algorithm revised as suggested
R1.8	✓	Stopping rule implemented as suggested
R1.9	✓	Mean squared errors provided as suggested
R1.10	✓	Pairwise heat maps provided as suggested
R1.11	✓	No actions required
R1.12	✓	Text revised as suggested
R2.1	✓	No actions required
R2.2	✓	Adaptive bandwidth implemented as suggested
R2.3	✓	Weights rescaled in the standard manner as suggested

R2.4	✓	Numerical results revised as suggested
R2.5	✓	Text revised as suggested
R2.6	✓	Text revised as suggested
R2.7	✓	Incorrect text removed as suggested
R2.8	✓	Additional text on summary statistics provided
R2.9	✓	Text revised as suggested
R2.10	✓	Text revised as suggested
R2.11	✓	Text revised as suggested
R2.12	✓	Figure title revised as suggested
R2.13	✓	Text revised as suggested
R2.14	✓	Text revised as suggested

Reviewer 1

A review of “Calibrating models of cancer invasion: parameter inference using Approximate

Bayesian Computation and gradient matching” by Y Xiao, L Thomas and MAJ Chaplain.

R1.1 This is an interesting and exciting piece of Mathematical Biology that has been a pleasure to read.

The authors review the construction of a Partial Differential Equation (PDE) model of cancer invasion, and they then proceed to fit parameters to this model. Parameter fitting is carried out with both an Approximate Bayesian Computation (ABC) and a gradient matching technique. The thrust of this manuscript is in demonstrating a successful parameter inference pipeline. This approach is likely to be emulated by other authors.

There are clear strengths to this expository work, and these are set out, section-by-section, in the following paragraphs. As this work introduces well-described ABC inference techniques to the Mathematical Biology literature, a number of directions for future work are

suggested for the author's consideration. Whilst this reviewer has considered the manuscript in its entirety, this feedback focuses on the ABC inference component of the work.

Thank-you for your helpful and constructive comments.

R1.2 Introduction. The authors start by explaining that “a common issue” that arises when using systems of differential equations to model various phenomena is that “some or all model parameters are not known”. They then say that model fitting techniques can provide the missing parameters. On a first reading, it appears that only point parameter estimates are required for the Mathematical Biology case study that is to follow. For many modelling applications, such as the study eloquently set out in the next section, this is perfectly sensible. Then, the authors raise the ambition of operating in a fully Bayesian setting -- setting priors on the parameters, observing experimental data, and then arriving at a posterior distribution to fully characterise the parameters. The authors might wish to explore the possibility of using the entire posterior distribution to mark the uncertainty in the parameters, for example, by considering the posterior predictive distribution. The authors might wish to highlight the circumstances where a fully Bayesian approach would be advantageous.

Thank-you for these comments. The reviewer is correct that our goal in this paper is to obtain reliable point parameter estimates, and we have added text in the Introduction to make this absolutely clear from the outset. In future work, we would like to explore methods for reliable uncertainty quantification; in this case, a fully (approximate) Bayesian approach would be advantageous, but only if measurement error (or some other form of stochasticity) is incorporated into the model (see Alahmadi et al. 2020 - now also cited in our paper). We have added text to clarify these points.

Actions: Text revised in the introduction, and in the Discussion.

Edits:

Page 2-3: ...Here, we develop two calibration schemes, one based on ABC and the other on two-stage gradient matching, and apply them to a PDE model of cancer invasion and metastasis. Our focus is on accurate estimation of model parameters when applied to data simulated from the model -- a necessary first step for reliable inference. We leave the two important real-world issues of uncertainty quantification and model assessment (goodness-of-fit) for future work.

Page 15-16: ... Lastly, our current ABC scheme was not designed to measure the uncertainties in the estimates -- no analytic methods exist and bootstrapping is computationally infeasible. In future work, observation errors may be introduced to the reference dataset so uncertainties in parameter estimates can be assessed using a full posterior inference under the Bayesian framework. However, we realize the incorporation of observation errors can be a challenging problem. Alahmadi et al. [45] have argued the posterior distributions derived from those ABC methods which fail to adequately model the measurement errors may not accurately reflect the epistemic uncertainties in parameter values. Hence it is necessary to model and incorporate the errors in the correct way when using ABC methods.

Page 17: ... We also side-stepped the important issue of uncertainty quantification. A real-world application will at least wish to consider quantifying uncertainty arising from measurement errors (see, e.g., [45]) and may also attempt to quantify other sources of uncertainty such as model mis-specification error.

R1.3 The authors might wish to make a small number of technical changes to this section. For example, the authors ought to formalise their references to “the best parameter values”, perhaps by explaining that they make use of Bayes estimators. Further details, such as the appropriate loss function the authors have considered, could be specified to ensure a complete description of their work.

Agreed. The final parameter estimates returned from the ABC scheme were the average parameter values obtained in the final round, in the fully Bayesian setting, this would be the “posterior mean” - a Bayes estimator with a corresponding quadratic loss function. In order to associate these issues with a better context, we address them in the Methods section, where the details of our ABC scheme were provided. The additional explanations are now added at the end of subsection 2.c)iii) “ABC-BCD (Approximate Bayesian Computation – Bhattacharyya distance) optimization scheme”.

Actions: Text revised in subsection 2.c)iii).

Edits: Page 8: ... (iv) After the final samples of all parameters are obtained, we take the means of these samples to be the estimated parameter values that can give the best fit to the synthetic data. (In the fully Bayesian setting, this would be the “posterior mean”, which is a Bayes estimator that has a quadratic loss function.)

R1.4 Methods. The PDE system is set out with stunning clarity and a standard non-dimensionalisation is carried out. The description of the ABC method could, perhaps, be edited to make full use of the opportunities offered by Bayesian inference. Specifically:

(a) Step (i) imposes a uniform prior on the parameters in the non-dimensional setting. Whilst a uniform prior is frequently encountered in this line of research, there are circumstances where alternative priors are better-suited. For example, there are occasions where it is necessary to use an informative prior.

The authors might wish to acknowledge that there are specific circumstances where it is necessary to first impose priors in the dimensional PDE setting, and then to translate these priors into the non-dimensional PDE formulation. This is a consequence of not all priors being invariant under a change of coordinates.

Good point, which we added in the discussion.

Action: Text revised in the Discussion.

Edits: Page 15: ... (iii) In the fully Bayesian setting, informative priors may be preferable under certain circumstances. When a differential equation model is involved, it can be more rigorous to propose the prior distributions in the dimensional setting and then translate them into non-dimensional ones, since not all the priors are invariant under the change of coordinate in some cases. Fortunately, this is not a concerning issue in our work. Our setting

is not completely Bayesian, the initial distributions of the parameters in this study only serve to influence the speed of convergence, they should not have an impact on the final point estimates, as long as the final point estimates are within the corresponding initial distributions.

R1.5 (b) The observant reader might recognise the similarities between the author's approach, and the ABC Sequential Monte Carlo (ABC-SMC) method contained in reference [15]. The authors might wish to include a few sentences discussing the similarities and differences.

Agreed. We included a few sentences to address the similarities and differences between our ABC scheme and the ABC-SMC method in Toni et al. (reference [15] of the manuscript) at the end of subsection 2.c)iii) "ABC-BCD (Approximate Bayesian Computation – Bhattacharyya distance) optimization scheme".

Action: Text revised in subsection 2.c)iii).

Edits: Page 9: ... Our ABC scheme has both similarities and differences to the Approximate Bayesian Computation - Sequential Monte Carlo (ABC-SMC) scheme proposed in Toni et al [15]. Both schemes use the idea of weight resampling and particle perturbations to derive the parameters to be evaluated in the next round from the ones being evaluated in the current round. On the other hand, the main differences between our scheme and the traditional ABC-SMC scheme are:

(i) The discrepancy between the summary statistics and the computation of weights are two separate processes in the traditional ABC-SMC scheme, while we combined them into one in our scheme. If t rounds are carried out in total, the traditional ABC-SMC scheme usually sets up a decreasing sequence of tolerance levels $\epsilon_1, \epsilon_2, \dots, \epsilon_t$ at the beginning. After round 1, the parameters to be evaluated in the next round are resampled and perturbed from the ones being evaluated in the current round using weights associated with parameter densities, the weights here act as a first filter. Then the tolerance levels are introduced as a second filter, the simulated and reference summary statistics are compared using certain discrepancy measuring metric, a perturbed parameter set is accepted if and only if the discrepancy between its simulated summary statistics and the reference ones is less than the tolerance level of the current round. While in our scheme, we incorporated the discrepancy measurements into the computation of weights, the weights are then used as the only filter to resample the parameters and ensure convergence..

(ii) Due to the absence of observation errors in our ABC application, a rigorous analysis of parameter densities was not undertaken. Therefore, the weights of parameter sets in our scheme solely depend on the discrepancies between summary statistics. While in the traditional ABC-SMC scheme, weights of parameter sets are usually related to parameter densities.

R1.6 (c) The authors do not seek the entire posterior distribution, but rather, they zoom in on a mean value. This has the advantage of providing a point estimate parameter for the model. For certain models this would mean the opportunity to fully characterise the uncertainty in the parameter-set is passed up, and a valid posterior density is not provided. This arises as the parameters are sampled according to a resampling probability (see part (iii)(a)(iv)) rather

than the prior distribution, and presumably not re-weighted accordingly. The authors are up-front about this: for example, in Figure 2, an “initial density” and a “final density” are shown.

Yes, our current ABC scheme is not designed for assessing uncertainties on the parameter estimates, it was built to retrieve the parameters which give the best fit to the reference data. We agreed the calculation of resampling probabilities needs to be improved, the weights are now rescaled in a standard manner (weights/sum of weights) to obtain resampling probabilities, which is often seen in traditional ABC methods. We hope the changes made in subsection 2.c)iii) “ABC-BCD (Approximate Bayesian Computation – Bhattacharyya distance) optimization scheme” give a better clarification on the issue of resampling probability.

Actions: Text revised in subsection 2.c)iii).

Edits: Page 8: ... (i) Convert the discrepancy values to resampling weights. First calculate $w_i^ = \rho^{-t}$ for $i = 1, \dots, K$, where t is a positive integer increased by 50% in every subsequent round. Then rescale the weights in the standard manner: $w_i = w_i^* / \sum_{i=1}^K w_i^*$. In some simulations, there are computational singularities resulting in undefined real values; in these cases, the corresponding weights are set to 0.*

R1.7 (d) The authors have computed resampling weights by first determining $\rho^{-0.5}$, and then linearly re-scaling. It is clear that the authors arrived at this choice after exhaustive investigations. To ensure that full credit is given to their findings, the authors might wish to provide a brief motivation for this choice (especially of the exponent shown above). It might also be worth mentioning that K can be j -dependent.

Thank-you for pointing this out. We have amended the algorithm to use adaptive bandwidths in weight calculations. Under such settings, parameter sets close to the reference values will be resampled with heavier weights in later rounds, convergence to the true values can then be guaranteed. The fact that K can be j -independent is also mentioned. We hope the adjustments made in subsection 2.c)iii) give a clear explanation of our points.

Actions: Text revised in subsection 2.c)iii).

Edits:

Page 7: ... (a) Sample K sets of parameter values from the initial distributions. Note that K and round indicator j are independent in this study.

Page 8: ... (i) Convert the discrepancy values to resampling weights. First calculate $w_i^ = \rho^{-t}$ for $i = 1, \dots, K$, where t is a positive integer increased by 50% in every subsequent round. Then rescale the weights in the standard manner: $w_i = w_i^* / \sum_{i=1}^K w_i^*$. In some simulations, there are computational singularities resulting in undefined real values; in these cases, the corresponding weights are set to 0.*

Page 8: ... The optimization scheme is stochastic, with the amount of Monte Carlo error controlled by the number of samples, ... The bandwidth of weights t was chosen to start from 0.5 and increased by 50% in every subsequent round, it was reset to 0.5 when we

proceeded to evaluate the next equation. By setting such stopping criterion and adaptive bandwidths for weights, the resampling surface became steeper in later rounds, parameter sets had minor Bhattacharyya distances to the reference values could then be resampled with heavier weights and convergence to the true values could be guaranteed.. For the perturbation value h in (iii)(d)iii, we chose $h = 0.05$. Lastly, as a further diagnostic of the scheme, we ran it on the two additional reference datasets that had been generated using different true parameter values (table S4).

R1.8 (e) The authors are right to acknowledge that a “stopping rule” has not been specified. A very simple option would be to plot the mean parameter as the algorithm unfolds, and to use this as a basis for a simple stopping rule. It is not necessary to address this in full in this manuscript, and a brief description of how a stopping rule could be implemented would probably satisfy most readers.

Good idea. We have now established the “stopping rule” in our scheme. We hope the additional text in subsection 2.c)iii) gives a good explanation of the stopping rule.

Actions: Text revised in subsection 2.c)iii).

Edits:

*Page 8: ...Calculate the average (mean) discrepancy of the parameters in the current round. Check if the distance (difference in absolute value) between the average discrepancy of the parameters in the current round and that of the parameters in round 1 has reached a threshold value, i.e. $|\rho_{y^i} - \underline{\rho}_{y^i}| \leq \epsilon * \underline{\rho}_{y^i}$, where ϵ is a positive number less than 1.*

*Page 8: ... Here we used $K = 10000$. We checked the Monte Carlo error by undertaking two additional runs on the same reference dataset and computing the standard error of the parameter estimates across the three runs. Overall accuracy is governed by the stopping criterion $|\rho_{y^i} - \underline{\rho}_{y^i}| \leq \epsilon * \underline{\rho}_{y^i}$ and the increasing bandwidth t in weight calculations. Different ϵ 's were used in the evaluations of the three different density profiles. It was set to be 0.8 when evaluating the ECM density profile alone, to prevent particle depletion in early rounds when only one parameter was being estimated. We then raised it to 0.9 in the evaluations of ECM and MDE profiles. Finally, it was set to be 0.98 when all density profiles are being evaluated. In our opinion, particle depletion is a concerning issue in early rounds, but should not cause any troubles in the last few rounds since our goal is to obtain accurate parameter point estimates. The bandwidth of weights t was chosen to start from 0.5 and increased by 50% in every subsequent round, it was reset to 0.5 when we proceeded to evaluate the next equation. By setting such stopping criterion and adaptive bandwidths for weights, the resampling surface became steeper in later rounds, parameter sets had minor Bhattacharyya distances to the reference values could then be resampled with heavier weights and convergence to the true values could be guaranteed.*

The reviewer has no specific comments concerning the gradient matching scheme set out in this section.

Results. This section carefully describes the outputs of the authors' investigations. In short, the authors picked specific parameters, generated in silico data, and then sought to recover

the parameters with the inference algorithm. To ensure that readers fully appreciate the benefits of ABC inference, the authors might wish to consider:

R1.9 (a) The difference between the picked and recovered parameters is referred to as a “percentage error”. This is a snap-shot of the algorithm’s performance at a specified parameter-set, and, in this case, in the absence of any measurement error. The reviewer would like to raise the possibility of considering the mean squared error of the outputs, so that the error can be naturally decomposed into bias and variance components.

Thank-you for pointing this out. The mean squared errors of the estimates obtained using our ABC scheme are now added in table 2 and table S5 (in the supplementary material) as extra columns.

Actions: Results revised in Table 2 and Table S5.

Edits: See Table 2 and Table S5 in the revised manuscript.

R1.10 (b) Only the marginal “final densities” are shown. An interested reader might wish to see pairwise heatmaps of “final densities” too.

Good idea, a figure of pairwise heat maps of final parameter estimates is now added to the manuscript in the Results section.

Actions: Additional figure of pairwise heat maps provided in section 3.

Edits:

Page 10: Plots of initial and final densities for the parameters under this scheme are shown in figure 2, a figure of pairwise heat maps of these final densities is shown in figure 3.

See Figure 3 in the revised manuscript.

R1.11 (c) It is unfortunate that, due to computational constraints, it has not been possible to demonstrate the performance of the ABC method on a noisy dataset. Building the computer code to carry out this research must have been a formidable task, and the reviewer is satisfied that every effort would have been made to ensure its efficiency.

Thank-you. This is something that we would like to undertake as a future task. Including measurement error also gives the opportunity to assess uncertainty using ABC methods.

R1.12 Discussion. This final section is refreshingly realistic in its description of the manuscript’s findings.

The authors are right to raise “the model selection problem” as the next frontier in mechanistic modelling. Whilst it would be unfair to ask the authors to detail their future research plans, it might be helpful if, by way of example, they are proposing to make use of Bayes factors or similar tools to conduct this analysis.

We sincerely thank the reviewer for the suggestions on future directions. We have now added a few sentences to explain some possible ideas that can be extended from the current work.

Actions: Text revised in the Discussion.

Edits: Page 15-16: ... Lastly, our current ABC scheme was not designed to measure the uncertainties in the estimates -- no analytic methods exist and bootstrapping is computationally infeasible. In future work, observation errors may be introduced to the reference dataset so uncertainties in parameter estimates can be assessed using a full posterior inference under the Bayesian framework. However, we realize the incorporation of observation errors can be a challenging problem. Alahmadi et al. [45] have argued the posterior distributions derived from those ABC methods which fail to adequately model the measurement errors may not accurately reflect the epistemic uncertainties in parameter values. Hence it is necessary to model and incorporate the errors in the correct way when using ABC methods.

Reviewer 2

Comments to the Author(s)

This paper investigates statistical inference for a PDE cancer model using approximate Bayesian computation (ABC) and gradient matching. It provides an interesting comparison of these methods on a challenging application.

I have one major concern about this paper: the correctness of the ABC algorithm used. Therefore I recommend major corrections. Otherwise the paper performs a good simulation study and is well written, so I have only a few other minor comments. Details of major and minor comments are below.

Also if the authors have time in future, it would be interesting to extend this study, especially to investigate how ABC performs in the case of observation error. Furthermore I wonder if ABC is the best algorithm to use for optimization when the simulator is deterministic (i.e. when there is no observation error). For instance one could use stochastic approximation optimization methods (of the Kiefer-Wolfowitz type). However exploration of these issues is beyond the scope of what's needed for this paper to be accepted.

Thank-you for these helpful comments. Yes, we are planning to investigate both the performance of ABC in the presence of observation error (for parameter estimation and for uncertainty quantification, as mentioned above) as well as other optimization methods - thanks for the suggestion of Kiefer-Wolfowitz-like methods.

Major comment

=====

R2.1 The ABC algorithm used has some non-standard features compared to standard ABC-SMC algorithm (for example, see the review article of Marin et al 2012

<https://doi.org/10.1007/s11222-011-9288-2>). One of these features seems like it could invalidate the correctness of the algorithm for this application (see first point below). The others seem problematic for general use of ABC, but not this particular application (see second point below).

In the fully Bayesian setting, the ABC methods aim at not only obtaining the parameter estimates but also drawing inference on the posterior distributions and assessing uncertainties. But in our application, due to the absence of observation errors, our algorithm was designed to retrieve the parameter estimates in the most efficient way rather than drawing a full inference on posterior distributions, hence there were some unusual features in our algorithm compared to the traditional ABC methods. We have made certain adjustments to the algorithm to ensure the correctness of the ABC application and focus on retrieving parameter point estimates at the same time. We hope the responses to the reviewer's later comments can clarify these adjustments.

ABC bandwidth

R2.2 The paper's ABC algorithm uses $\text{weight} = \text{distance}^{-1/2}$. ABC-SMC algorithms would typically include a bandwidth variable h here e.g. $\text{weight} = \text{distance}^{-1/2h}$ which is reduced as the iterations of the algorithm progress (early papers used a pre-specified schedule of values, but reducing h adaptively is now more common). This is necessary for the algorithm to converge on the true parameter values.

I suggest using a reducing bandwidth here and rerunning the simulations. Alternatively the authors could explain why they wish to use a fixed bandwidth, and adjust this discussion accordingly i.e. note that the algorithm does not converge on the true parameters, but instead gives approximate results, and justify why this is good enough for the current purposes. However in the latter case, I think the issues in the next point become more important.

Thank-you for pointing this out. We have amended the algorithm to use adaptive bandwidths in weight calculations. Under such settings, parameter sets that have smaller distances to the true values will be resampled with heavier weights in later rounds and convergence to the true parameter values can then be guaranteed. We hope the highlighted text at the beginning of page 8 of the revised manuscript gives a clear explanation of the adaptive bandwidth.

Actions: Text revised in section 2.c)iii)

Edits:

Page 8: ... (i) Convert the discrepancy values to resampling weights. First calculate $w_i^ \equiv \rho^{-t}$ for $i = 1, \dots, K$, where t is a positive integer increase by 50% in every subsequent round. Then rescale the weights in the standard manner: $w_i = w_i^* / \sum_{i=1}^K w_i^*$. In some simulations, there are computational singularities resulting in undefined real values; in these cases, the corresponding weights are set to 0.*

Page 9: ... The bandwidth of weights t was chosen to start from 0.5 and increased by 50% in every subsequent round, it was reset to 0.5 when we moved on to evaluate the next equation.

Other issues with algorithm

R2.3 Some other unusual features of the ABC algorithm are:

- * The weights do not include a term for the density of the proposed parameters.
- * Weights are not rescaled by the standard formula (dividing by the sum of the weights).

This means that the usual sequential Monte Carlo / population Monte Carlo analysis of the algorithm does not hold and so this algorithm does not output samples from the usual ABC approximation to the posterior density. It is somewhat unclear therefore what distribution the algorithm samples from. In general it would be preferable to have some reassurance that this algorithm does not have some undesirable properties compared to standard ABC e.g. large extra approximation error, or no guarantee of convergence.

However for their particular application, the authors simply wish to use ABC as an optimisation algorithm. Thus, so long as it finds parameter values which approximately minimise the distance, then this is sufficient for the desired behaviour. (However to do so, the bandwidth issue above probably needs to be addressed.)

Thank-you for pointing this out. Due to the absence of measurement errors in our ABC application, we discarded the idea of drawing a full posterior inference on parameter densities. Therefore, we did not consider parameter densities in the present work. We will mention "the analysis of parameter densities will be included in our future work if measurement errors are available and a full posterior inference will be undertaken under the Bayesian framework". Secondly, the weights are now rescaled in the standard manner as suggested by the reviewer. We hope the revised algorithm in subsection 2.c)iii) of the manuscript explains our adjustments well enough.

Actions: Text revised in subsection 2.c)iii) and the Discussion.

Edits:

Page 8: ... (i) Convert the discrepancy values to resampling weights. First calculate $w_i^ \equiv \rho^{-t}$ for $i = 1, \dots, K$, where t is a positive integer increase by 50% in every subsequent round. Then rescale the weights in the standard manner: $w_i \equiv w_i^* / \sum_{i=1}^K w_i^*$. In some simulations, there are computational singularities resulting in undefined real values; in these cases, the corresponding weights are set to 0.*

Page 15-16: ... Lastly, our current ABC scheme was not designed to measure the uncertainties in the estimates -- no analytic methods exist and bootstrapping is computationally infeasible. In future work, observation errors may be introduced to the reference dataset so uncertainties in parameter estimates can be assessed using a full

posterior inference under the Bayesian framework. However, we realize the incorporation of observation errors can be a challenging problem. Alahmadi et al. [45] have argued the posterior distributions derived from those ABC methods which fail to adequately model the measurement errors may not accurately reflect the epistemic uncertainties in parameter values. Hence it is necessary to model and incorporate the errors in the correct way when using ABC methods.

Minor comments

=====

R2.4 1. It would be preferable to show numerical results using mathematical notation e.g. 6.9×10^6 rather than 6.9e6. This is particularly useful for numbers which can simply be represented as decimals e.g. 0.012 is much easier to interpret in a table than 1.2e-2.

Agreed. Mathematical notations are now implemented throughout the numerical results in the manuscript.

Actions: Numerical values revised throughout the manuscript.

Edits: See the updated results in the tables of the revised manuscript.

R2.5 2. It might be worth briefly discussing the recent paper "A comparison of approximate versus exact techniques for Bayesian parameter inference in nonlinear ordinary differential equation models" by Alahmadi et al (<https://doi.org/10.1098/rsos.191315>). They discuss how ABC methods can fail for differential equation models if badly implemented. Although this problem applies mainly to the case of observation error, it might be interesting to point this out to readers of your paper.

Thank you for pointing this out. We have come across this article in our literature review; it gives a good explanation of the difficulties one may encounter when applying ABC methods to noisy data generated from differential equation models. When the observation errors are introduced, the inference may become more challenging. We hope the additional text given in the fourth paragraph of the discussion covers this issue.

Actions: Text revised in the Discussion.

Edits: Page 15-16: ...Lastly, our current ABC scheme was not designed to measure the uncertainties in the estimates -- no analytic methods exist and bootstrapping is computationally infeasible. In future work, observation errors may be introduced to the reference dataset so uncertainties in parameter estimates can be assessed using a full posterior inference under the Bayesian framework. However, we realize the incorporation of observation errors can be a challenging problem. Alahmadi et al. [45] have argued the posterior distributions derived from those ABC methods which fail to adequately model the measurement errors may not accurately reflect the epistemic uncertainties in parameter values. Hence it is necessary to model and incorporate the errors in the correct way when using ABC methods.

R2.6 3. The authors say in a few places that ABC is not suitable for assessing uncertainty in the parameter estimates. Could the authors expand on what type of uncertainty they wish to

capture here? From a Bayesian or maximum likelihood viewpoint, if the likelihood is a spike at a single parameter value then there is no uncertainty! Other sources of uncertainty exist, such as model misspecification, but are quite hard to capture by any method.

Our comment about ABC not being suitable to assess uncertainty was intended to apply only to our scheme, and we have clarified this in the text. We have also added an explicit reference to both observation uncertainty and model mis-specification error at the end of the discussion.

Actions: Text revised in Methods and Discussion.

Edits:

Page 6: Hence we consider our scheme to be a stochastic optimization scheme, suitable for estimating parameter values but not for assessing uncertainty on the estimates."

Page 17: (penultimate paragraph in Discussion) ... We also side-stepped the important issue of uncertainty quantification. A real-world application will at least wish to consider quantifying uncertainty arising from measurement errors (see, e.g., [45]) and may also attempt to quantify other sources of uncertainty such as model mis-specification error.

R2.7 4. Page 2 says "Under the assumption that the differential equation model is correct and that the observations arise independently from a normal distribution then the least squares parameter estimates are maximum likelihood estimates." I think it is not necessary for the differential equation model to be correct for the least squares estimates and MLE to coincide.

Thanks - corrected.

Actions: incorrect clause removed.

Edits: Page 2: Under the assumption that the observations arise independently from a normal distribution then the least-squares parameter estimates are maximum likelihood estimates.

R2.8 5. Page 6 says "the algorithm will eventually converge on the best fitting values with no variation between parameter sets". As well as the bandwidth issue mentioned above, another issue with this is the use of summary statistics. There's an implicit assumption here that an exact match of summary statistics implies an exact match of the full data. This should be mentioned, with a brief discussion of how reasonable this is for the application under investigation.

Thank-you for raising this issue. We have now added some additional explanations of summary statistics in subsection 2.c)ii); we hope they address this issue properly.

Actions: Text revised in subsection 2.c)ii)

Edits: Page 6-7: ... For our scheme to converge to the true parameter values, there is an implicit assumption that a Bhattacharyya distance of zero means that the data perfectly match model predictions. This assumption would fail if it were possible for the model to generate a dataset with the same means and variances as the data, but from a different set of parameter values. This is unlikely in our case study; in a real-world application, where the model is necessarily an approximation to the real-world data-generating process and the data also contain measurement errors, it would be required that smaller values of the discrepancy measure correspond to better values of the model parameters.

R2.9 6. Page 9. Mention which optimization algorithm was used when using the optim function in R.

Thanks for addressing this. The default method "Nelder-Mead" was used when calling the optim function. We have now mentioned this on page 10 of the manuscript.

Actions: Text revised in subsection 2.d)

Edits: Page 10: ...the estimated parameter values which gave the best fit to the noisy data were obtained by minimizing (using the optim function with the default "Nelder-Mead" method in R) the objective function:

R2.10 7. Page 11 says "Constraining certain parameters when the gradients are inaccurate will make the estimates even worse, since this makes the optimization scheme loses its ability to adjust itself." Can you give some more explanation or reword here? It's not obvious to me that this is true or exactly what "the ability to adjust itself" means.

Apologies for not making this clear enough. We have rephrased the original sentence and hope the readability is improved now.

Actions: Text revised in subsection 3.b)

Edits: Page 13: ... Constraining certain parameters when the gradients are inaccurate will make the estimates even worse since only a subset of the parameters can be altered to minimize the value of the objective function, this makes the optimization scheme loses its flexibility to a certain extent, which may affect the parameter estimates in a negative way.

R2.11 8. Page 12 says "If we look at the PDE model itself, the parameters in the tumour cells equation are all associated with numerically complex terms, and thus the information hidden behind the PDE model that can help to improve the accuracy of their estimates is therefore limited." Reword this? Currently it seems to say that numerical complexity implies lack of informativeness, which seems incorrect.

Agreed. We have now rephrased the original sentence.

Actions: Text revised in the second bullet point of the second paragraph of Discussion.

Edits: Page 15: ... If we look at the PDE model itself, the parameters in the tumour cells equation are all associated with numerically complex terms, it can be harder to acquire the information hidden behind the PDE model that can help to improve the accuracy of their estimates.

R2.12 9. In Figure 4, on page 13, it's hard to make out different lines. This is better in the larger sized versions of these plots at the end of the document. Maybe mention that these are available in the caption to Figure 4? Also, the ^ character seems to be misplaced in several of the plot headings.

We have now changed the plots to larger sizes and mentioned the larger sized versions are also provided. The “^” symbol in the plot headings means the gradients being presented in the plots are estimated values predicted by the GAM, not exact values calculated from the finite difference scheme.

Actions: Plots in Figure 4 were changed to a larger size in the manuscript, text revised in the heading of Figure 4.

Edits: Page 14: ... Figure 4: Temporal and spatial gradients in the model at different measurement error CVs. Red dashed lines: reference gradients predicted by GAM at CV equals 0. Dark purple solid lines: true gradients calculated by the finite difference scheme. Other solid lines show estimated gradients averaged over time and 200 data sets from the gradient matching scheme at CVs of 0.01 (black), 0.025 (green), 0.05 (blue), 0.075 (cyan) and 0.1 (magenta). A larger-sized version of these plots is provided in the Github repository mentioned in the “Data Accessibility” statement at the end of the manuscript.

R2.13 10. Page 13 says "However, this difference here should be accounted for the errors in the tumour cell-related estimates when no perturbation was added." I didn't understand what this sentence meant. Is it possible to explain further, or is there a typo in the sentence?

Apologies for not making this sentence clear enough. We hope the revised text in the manuscript explains our point better.

Actions: Text revised in the Discussion.

Edits: Page 16: ... since all the gradients in our scheme were predicted by GAM, true gradients calculated by the finite difference scheme were not used. However, the difference between the GAM predicted and true gradients should be accounted for the errors in the tumour cell-related estimates when no perturbation was added.

R2.14 11. Page 15 says "Overall, we believe the parameter estimates for PDE models using statistical approaches is a strong alternative to searching high-dimensional parameter space by hand-tuning". What is meant by "hand-tuning" here? I couldn't find this phrase anywhere earlier in the paper.

Hand-tuning means changing the parameter values and substituting them back to the numerical solver manually. We have revised this sentence in the manuscript.

Actions: Text revised in the Discussion.

Edits: Page 17: ...Overall, we believe the parameter estimates for PDE models using statistical approaches is a strong alternative to searching high-dimensional parameter space manually, which requires a very powerful numerical scheme if the differential equations system is complicated.

Possible typos

=====

1. Page 6: "suitable for estimating parameter values but ****not not**** for assessing uncertainty on the estimates"

Fixed.

2. Page 8: "The discrepancies between the two sides of the equations ****was**** then calculated"

Fixed.

3. Page 10: "but in general, they were quite insensitive to the increase in measurement errors, except for $\partial m/\partial t$, the deviations at the left tail seem to increase with the CV." Is there a word missing somewhere here e.g. should it be "****where**** the deviations...?"

Fixed.

4. Page 11: "The complex spatial gradients (e.g. second order spatial derivatives, haptotaxis term) also ****shown**** obvious deviations"

Fixed.

Bibliography:

1. Alahmadi A, Flegg J, Cochrane D, Drovandi C, Keith J. 2020 A comparison of approximate versus exact techniques for Bayesian parameter inference in nonlinear ordinary differential equation models. *R. Soc. open sci.* 7.
2. Toni T, Welch D, Strelkowa N, Ipsen A, Stumpf P. 2008 Approximate Bayesian Computation scheme for parameter inference and model selection in dynamical systems. *Interface. Focus.*6,187–202.

Appendix C

Response to the decision letter received on 18/05/2021

Dear Professor Rogers,

Thank you for the decision of accepting our revised manuscript, and for the positive comments and suggestions provided by the two reviewers. Below we reproduce each comment (in regular font) and how we dealt with it (in *red italic font, with underlining used to indicate text changes in the manuscript*). Once again, thank you for accepting our manuscript and we look forward to producing better research work in the future.

With best wishes,

Yunchen Xiao and coauthors

Comments	Status	Actions
E.f.1	✓	No actions required
R2.f.1	✓	Text revised as suggested
R.2.f.2	✓	Text revised as suggested

Associate Editor Comments to Author (Professor Tim Rogers):

Associate Editor: 1

Comments to the Author:

E.f.1 The final revisions suggested by the referee are optional.

Thank-you. We have made the final minor revisions as suggested by the referees, our responses to their comments are attached below.

Reviewer comments to Author:

Reviewer: 2

Comments to the Author(s)

The authors have addressed all the points from my review and I am happy to recommend that the paper is accepted for publication.

I have two minor comments which the authors could perhaps incorporate in their final draft. These are listed below using the numbering scheme from the authors' response.

We would like to express our gratitude to the reviewer for recommending our revised manuscript for publication. Our responses to the reviewer's two minor comments are presented below.

R2.f.1 R2.2 It might be worth mentioning that there are some differences between your algorithm and standard ABC-SMC, so that readers are aware that some modifications are needed if they are interested in inferring the full posterior.

Thank-you for mentioning this. We have discussed the similarities and differences between our algorithm and standard ABC-SMC in section 2.c)iii), page 9 of our revised manuscript. We have made some further modifications to make this even clearer.

Actions: Text revised in section 2.c)iii) (with relevant changes underlined).

Edits: Page 9: ... Our ABC scheme has both similarities with and differences from the Approximate Bayesian Computation - Sequential Monte Carlo (ABC-SMC) scheme described in Toni et al [15]. Both schemes use the idea of weighted resampling and particle perturbations to derive the parameters to be evaluated in the next round from the ones being evaluated in the current round. The main differences between our scheme and the traditional ABC-SMC scheme are:

(i) The discrepancy between the summary statistics and the computation of weights are two separate processes in the ABC-SMC scheme, while we combined them into one in our scheme. If t rounds are carried out in total, the ABC-SMC scheme usually sets up a decreasing sequence of tolerance levels $\epsilon_1, \epsilon_2, \dots, \epsilon_t$ at the beginning. After round 1, the parameters to be evaluated in the next round are resampled and perturbed from the ones being evaluated in the current round using weights associated with parameter densities; the weights here act as a first filter. Then the tolerance levels are introduced as a second filter: the simulated and reference summary statistics are compared using a discrepancy measuring metric, and a perturbed parameter set is accepted only if the discrepancy between its simulated summary statistics and the reference ones is less than the tolerance level of the current round. By contrast, in our scheme, we incorporated the discrepancy measurements into the computation of weights; the weights are then used as the only filter to resample the parameters and ensure convergence..

(ii) Due to the absence of observation errors in our ABC application, a rigorous analysis of parameter densities was not undertaken. Therefore, the weights of parameter sets in our scheme solely depend on the discrepancies between summary statistics. While in the ABC-SMC scheme, weights of parameter sets are usually related to parameter densities and help to derive the joint and marginal posterior densities. Thus, for readers who wish to draw posterior inferences, it is necessary to incorporate observation errors in the step of data simulation and parameter densities in the step of weight calculations.

R2.f.2 R2.13 This sentence is clearer now: "However, the difference between the GAM predicted and true gradients should be accounted for the errors in the tumour cell-related estimates when no perturbation was added." But I am still unsure of what is meant by "should be accounted for". Does this mean it should be adjusted somehow?

Fair point, our text was confusing. We meant that it is the cause of the difference. We have revised the sentence in the manuscript.

Actions: Text revised in the Discussion.

Edits: Page 16: ... The difference between the GAM-estimated and true gradients is responsible for the errors in the tumour cell-related estimates when no perturbation was added.

Reviewer: 1

Comments to the Author(s)

The authors have substantially addressed every concern that I raised in my initial report. Thus, I am of the view that this manuscript is now suitable for publication in Royal Society Open Science.

We would like to thank the reviewer for recommending our revised manuscript for publication.